# STOCHASTIC BRIDGES AS EFFECTIVE REGULARIZERS FOR PARAMETER-EFFICIENT TUNING

## ABSTRACT

Parameter-efficient tuning methods (PETs) have achieved promising results in tuning large pre-trained language models (PLMs). By formalizing frozen PLMs and additional tunable parameters as systems and controls respectively, PETs can be theoretically grounded to optimal control and further viewed as optimizing the terminal cost and running cost in the optimal control literature. Despite the elegance of this theoretical grounding, in practice, existing PETs often ignore the running cost and only optimize the terminal cost, i.e., focus on optimizing the loss function of the output state, regardless of the running cost that depends on the intermediate states. Since it is non-trivial to directly model the intermediate states and design a running cost function, we propose to use latent stochastic bridges to regularize the intermediate states and use the regularization as the running cost of PETs. As the first work to propose regularized PETs that use stochastic bridges as the regularizers (running costs) for the intermediate states, we show the effectiveness and generality of this regularization across different tasks, PLMs and PETs. In view of the great potential and capacity, we believe more sophisticated regularizers can be designed for PETs and better performance can be achieved in the future.

## 1 INTRODUCTION

Recent years have witnessed the dramatic growth of pre-trained language models (PLMs) in various fields (Devlin et al., 2019; Dosovitskiy et al., 2021). As the size of PLMs continues to increase, the number of parameters has now even reached hundreds of billions (Brown et al., 2020; Smith et al., 2022), making fine-tuning the whole PLM both computationally impractical and environmentally unfriendly. In view of this, a variety of Parameter-Efficient Tuning methods (PETs) are proposed (Houlsby et al., 2019; Hu et al., 2022; Zaken et al., 2022; Lester et al., 2021). By only tuning a small number of additional parameters, PETs can be comparable to full-parameter fine-tuning.

Despite the success of PETs, their underlying mechanism remains an open problem. Recently, several works have proposed to interpret PETs with optimal control theory. Yang & Liu (2022) first show that the optimization in Prefix Tuning (Li & Liang, 2021) (a typical method of PETs) can be considered as the search for optimal control variables in the context of optimal control, i.e., the trainable prefixes can be seen as the control variables that drive the PLM (the system) to the desired output. Ding et al. (2022) further show that the optimal control perspective can be applied to almost all PETs. The optimization of PETs' parameters can be seen as minimizing the two cost functions in the optimal control literature: (1) *terminal cost* $\mathcal{L}_T$, which measures the quality of the terminal state, and (2) *running cost* $\mathcal{L}_R$, which measures the feasibility of the controlled intermediate states and the control variables. Although $\mathcal{L}_T$ can well correspond to the loss function of the model output, $\mathcal{L}_R$ is only vaguely described as the regularizers on the parameters of PETs (control variables) in Yang & Liu (2022) and Ding et al. (2022), *ignoring the dependency of $\mathcal{L}_R$ on the intermediate states*.

In this work, we show that designing a running cost to regularize intermediate states not only makes the optimal control perspective of PETs more theoretically sound, but also empirically leads to better PETs. We begin by assuming that in PLMs, the intermediate hidden states for generating different tokens in a sentence have different dynamics (or trajectories), and the dynamics can be approximated with stochastic processes in a latent space. Specifically, we first freeze the PLM and learn a mapping from the original hidden state space of the PLM to a latent space. In the latent space, the dynamics of the intermediate hidden states for generating different target tokens can be approximated with

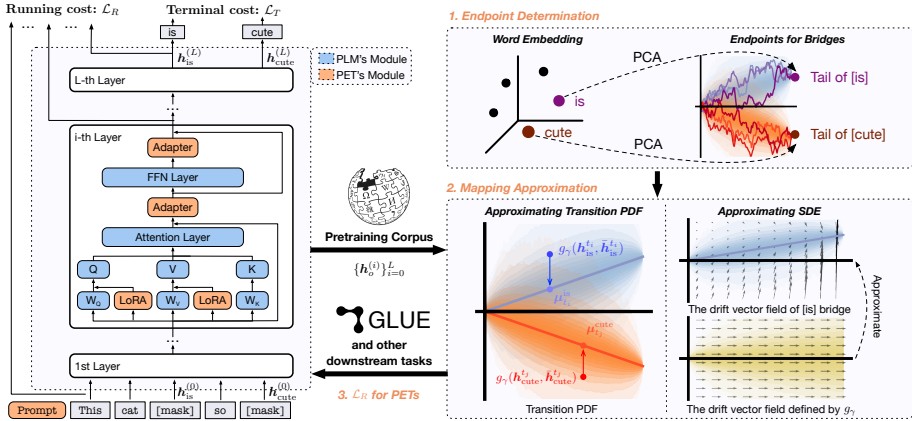

Figure 1: An overview of our proposed latent stochastic bridge regularizer.

different target-specific *diffusion bridges*. The obtained mapping can then be plugged to the model to regularize the intermediate hidden states when training PETs parameters. Besides, since a diffusion bridge is (1) a Markov process and (2) a solution to a stochastic differential equation (SDE), we correspondingly propose two methods to learn the mapping: (1) fitting the Markov transition probability density function (PDF) and (2) fitting the SDE directly. These two methods act as a trade-off between efficiency and effectiveness: the first method incurs only negligible computational cost and has satisfactory results, while the second one is slower, but yield better regularizers.

We conduct experiments on different PLMs of different sizes, and the experimental results on GLUE (Wang et al., 2019) under both full-set and few-shot settings demonstrate the effectiveness of our proposal across four different PETs. Further analyses show that the learned regularizer helps pull apart the hidden states of different label words. We also observe that when we project the intermediate hidden states of PETs without our regularizer into our latent space, the better the PETs perform, the closer the latent states are to our latent bridges. This spontaneous approaching behavior may indicate that stochastic-bridge-like latent dynamics naturally exists in well-trained PETs.

In summary, our work has the following contributions: (1) Guided by the perspective of optimal control for PETs, we design latent stochastic bridge regularizers on the intermediate states during the training of PETs. (2) We propose two methods to construct the latent space according to the two representations of stochastic bridges, offering a trade-off between efficiency and effectiveness. (3) Our regularizers are shown to be effective and general across different PLMs, different PETs, and different tasks. (4) We show that well-trained PETs without any regularization spontaneously exhibit stochastic-bridge-like latent dynamics.

## 2 BACKGROUND

### 2.1 DEFINITION AND MATHEMATICAL NOTATIONS

Consider using a $L$-layer PLM with the vocabulary $\mathbb{V}$ to handle a text-to-text task $\mathcal{D}$. For each sample $(\boldsymbol{x}, y) \in \mathcal{D}$, $y \in \mathbb{V}$ is the output token and $\boldsymbol{x} \in \mathbb{V}^N$ is the input token sequence [1], where $N$ is the length of $\boldsymbol{x}$. With $\boldsymbol{x}$ as the input, each layer of the PLM will output a sequence of hidden states, and we denote the hidden states of the $i$-th PLM layer as $\boldsymbol{h}^{(i)} = \{\boldsymbol{h}_j^{(i)}\}_{j=1}^N$, where $\boldsymbol{h}_j^{(i)} \in \mathbb{R}^d$ is the state at the position $j$ of the $i$-th layer, and $d$ is the model dimension. We denote the position where the model outputs the target $y$ as $o$, i.e., the model should predict $y$ with the hidden states $\boldsymbol{h}_o^{(L)}$.

### 2.2 OPTIMAL CONTROL PERSPECTIVE OF PETS

Conventionally, adapting the PLM to $\mathcal{D}$ requires full-parameter fine-tuning, which is given as:

$$\min_{\Delta\theta} \mathbb{E}_{\boldsymbol{x}, y \sim \mathcal{D}} \Big[ \mathcal{L}(\boldsymbol{h}_o^{(L)}, y) + \mathcal{R}(\Delta\theta) \Big], \quad \boldsymbol{h}^{(i)} = \begin{cases} \boldsymbol{h}^{(i-1)} + \mathcal{G}_{\theta+\Delta\theta}^{(i)}(\boldsymbol{h}^{(i-1)}), & i = 1, \dots, L, \\ \texttt{Embed}(\boldsymbol{x}), & i = 0, \end{cases} \quad (1)$$

---

[1] Here we assume $y \in \mathbb{V}$ since a sample where $\boldsymbol{y} \in \mathbb{V}^M$ can be decomposed to $M$ samples, The $i$-th sample is $([\boldsymbol{x}; \boldsymbol{y}_{<i}], y_i)$ for auto-regressive language modeling or $([\boldsymbol{x}; \boldsymbol{y}_{-i}], y_i)$ for auto-encoding language modeling.

where $\theta$ is the PLM parameters, $\Delta\theta$ is the full-parameter update, $\mathcal{L}(\cdot,\cdot)$ is the loss function, $\mathcal{R}(\cdot)$ is the regularization function, $\mathcal{G}_{\theta+\Delta\theta}^{(i)}(\cdot)$ is the forward propagation of the $i$-th PLM layer after updating the parameters, Embed transforms the input token sequence into input embeddings.

As $|\theta|$ continues to increase, full-parameter fine-tuning becomes impractical, and various PETs are proposed to mitigate this problem. Let $\phi = \{\phi^{(i)}\}_{i=0}^{L}$ be PETs' parameters. Ding et al. (2022) give a unified view of PETs from the perspective of optimal control, and Eq. 1 can be re-written as

$$\min_{\phi} \mathbb{E}_{\boldsymbol{x},y\sim\mathcal{D}}\Big[\mathcal{L}_T\big(\boldsymbol{h}_o^{(L)},y\big)+\sum_{i=0}^{L}\mathcal{L}_R\big(\phi^{(i)}\big)\Big], \quad \boldsymbol{h}^{(i)} = \left\{ \begin{array}{ll} \boldsymbol{h}^{(i-1)} + \tilde{\mathcal{G}}_\theta^{(i)}\big(\boldsymbol{h}^{(i-1)},\phi^{(i)}\big), & i=1,\ldots,L, \\ \big[\phi^{(0)}; \text{Embed}(\boldsymbol{x})\big], & i=0, \end{array} \right. \quad (2)$$

where $\tilde{\mathcal{G}}_\theta^{(i)}(\cdot,\phi^{(i)})$ represents the forward propagation of the $i$-th PLM layer intervened by PETs, $[\cdot;\cdot]$ is the concatenation operation, $\mathcal{L}_T(\cdot,\cdot)$ is the terminal cost and $\mathcal{L}_R(\cdot)$ is the running cost. Since $|\phi| \ll |\theta|$, PETs can greatly reduce the tuning cost (more details in Appendix A).

Typically for classification and generation tasks, $\mathcal{L}_T$ corresponds to the cross-entropy loss of the prediction, and $\mathcal{L}_R$ can be seen as a regularizer on PETs' parameters $\phi$. However, in the optimal control literature, $\mathcal{L}_R$ depends on not only the control variables $\phi$, but also the controlled intermediate states $\{\boldsymbol{h}_o^{(i)}\}_{i=1}^{L}$. In this paper, we show that including intermediate states to build $\mathcal{L}_R$ (i.e., using $\sum_{i=0}^{L}\mathcal{L}_R\big(\phi^{(i)},\boldsymbol{h}_o^{(i)}\big)$ instead of $\sum_{i=0}^{L}\mathcal{L}_R\big(\phi^{(i)}\big)$) not only makes the optimal control perspective of PETs more theoretically sound, but also empirically leads to better PETs.

## 2.3 Diffusion Bridges

A diffusion process $X = (X_t)_{t\in[0,T]}$ is a continuous-time Markov process. For any $t_a < t_b$, the diffusion process is equipped with a transition Probability Density Function (PDF) $p(t_b, b \mid t_a, a)$, which gives the probability density of reaching $b$ at time $t_b$ given the history of reaching $a$ at time $t_a$. A diffusion process is also the solution to an Itô SDE $d\tilde{X}_t = \mu(t,\tilde{X}_t)dt + \sigma(t,\tilde{X}_t)dB_t$, where $B_t$ is a standard Brownian motion, $\mu(\cdot,\cdot)$ is called drift function and $\sigma(\cdot,\cdot)$ is called diffusion function.

A diffusion bridge $X^{T;\alpha,\beta}$ is a diffusion process conditioning on the path observations of the two endpoints $(0,\alpha)$ and $(T,\beta)$, i.e., $X_0^{T;\alpha,\beta} = \alpha$ and $X_T^{T;\alpha,\beta} = \beta$. For simplicity, we assume $\alpha = 0$ in this work, and omit the superscript $\alpha$. We consider two typical diffusion bridges, the Brownian bridge and the Ornstein-Uhlenbeck bridge (OU bridge). We present here the properties of the Brownian bridge and leave the properties of OU bridge to Appendix B.

**Proposition 2.1** (The properties of the Brownian Bridge). A Brownian bridge $X^{T;\beta}$ with $X_0^{T;\beta} = 0$ and $X_T^{T;\beta} = \beta$ is the solution to the following SDE:

$$d\tilde{X}_t = (\beta - \tilde{X}_t)/(T-t)\, dt + dB_t, \quad \tilde{X}_0 = 0. \tag{3}$$

The transition PDF from $X_0^{T;\beta} = 0$ to $X_{t_b}^{T;\beta} = b$ is given as

$$p^{T;\beta}(t_b, b \mid 0, 0) = \frac{1}{\sqrt{2\pi t_b(T-t_b)}} \exp\Big[-\frac{(b-(t_b/T)\beta)^2}{2t_b(T-t_b)}\Big]. \tag{4}$$

Diffusion bridges and SDEs are battle-tested tools to describe the stochastic dynamics of complex systems in engineering (Sobczyk, 2013), finance (Wang & Sloan, 2011), biology (Horsthemke & Lefever, 1984), etc. Considering the dynamics of PLMs' hidden states are necessarily complex, diffusion bridges and SDEs serve as ideal tools for us to model the dynamics.

# 3 Latent Stochastic Bridges Regularizer

## 3.1 The Overall Framework

**Building latent dynamics in the latent space.** Since directly regularizing the intermediate states and constructing the running cost are non-trivial, we introduce a projection from the intermediate state space to a latent space, and leverage diffusion bridges as regularizers to construct the running cost. Specifically, we define a $r$-dimensional latent space $\mathbb{U} \subseteq \mathbb{R}^r (r < d)$ and a learnable mapping $g_\gamma : \mathbb{R}^d \times \mathbb{R}^d \to \mathbb{U}$, where $\gamma$ denotes the parameters. $g_\gamma$ projects the hidden state $\boldsymbol{h}_o^{(i)}$ and its context

state $\bar{\boldsymbol{h}}^{(i)}$ into the latent space $\mathbb{U}$ at each layer of the PLM. Since $\boldsymbol{h}_o^{(i)}$ is contextualized while latent bridges are not, introducing the dependency on $\bar{\boldsymbol{h}}^{(i)}$ can inform $g_\gamma$ about the context at the $i$-th layer and allow $g_\gamma$ to decontextualize the hidden states. We simply take the averaged states at the $i$-th layer $\bar{\boldsymbol{h}}^{(i)} = \frac{1}{N} \sum_{j=1}^N \boldsymbol{h}_j^{(i)}$ as the context. We define the latent states with discrete time in $\mathbb{U}$ as

$$\boldsymbol{u}_D(g_\gamma, \{\boldsymbol{h}_o^{(i)}\}_{i=0}^L) = \{t_{i+1}, g_\gamma(\boldsymbol{h}_o^{(i)}, \bar{\boldsymbol{h}}^{(i)})\}_{i=0}^L, \quad t_{i+1} = (i+1)/(L+2), \tag{5}$$

where $t_{i+1}$ is the normalized layer index. We include the 0-th layer (input layer) because some PETs (e.g., prompt tuning) act on the 0-th layer. We use $t_0 = 0, t_{L+2} = 1$ represent the two endpoints. By using natural cubic spline knotted at $\{\boldsymbol{h}_o^{(i)}\}_{i=0}^L$ to interpolate over $[-1, L+1]$, we further give a continuous representation of the states in the latent space $\mathbb{U}$ as

$$\boldsymbol{u}_C(g_\gamma, \{\boldsymbol{h}_o^{(x)}\}_{x\in[-1,L+1]}) = \{t_{x+1}, g_\gamma(\boldsymbol{h}_o^{(x)}, \bar{\boldsymbol{h}}^{(x)})\}_{x\in[-1,L+1]}, \quad t_{x+1} = (x+1)/(L+2) \in [0,1]. \tag{6}$$

**Learning the mapping from hidden state space to latent space.** Since adapting PLMs to downstream tasks can be seen as transferring the knowledge obtained from pre-training tasks to downstream tasks, we argue that the latent dynamics of intermediate hidden states for generating the same token $y$ should be similar in both the pre-training and downstream tasks. Therefore, we train the mapping $g_\gamma$ on the corpus that is used to pre-train the backbone PLM, and then apply the learned mapping to downstream tasks to encourage the latent dynamics to be similar to that in pre-training.

Specifically, we assume that the states to generate the token $y$ in the latent space $\mathbb{U}$ form a trajectory that is a path sampled from $X^{1;\beta_y}$ with high probability, where $X^{1;\beta_y}$ is the pre-determined diffusion bridge describing the latent dynamics to generate $y$, and $\beta_y$ is the tail endpoint of the diffusion bridge. More details of $X^{1;\beta_y}$ will be discussed in Section 3.2.

On the corpus where the PLM is pre-trained, we fix the PLM and use its hidden states $\{\boldsymbol{h}_o^{(i)}\}_{i=1}^L$ to learn $g_\gamma$ by maximizing the goodness of approximation for latent states $\boldsymbol{u}$ under the bridge $X^{1;\beta_y}$:

$$\gamma \leftarrow \arg\max_{\gamma'} \left[\text{goodness-of-approx}\big(\boldsymbol{u}(g_{\gamma'}, \{\boldsymbol{h}_o^{(\cdot)}\}), X^{1;\beta_y}\big)\right], \tag{7}$$

where $\boldsymbol{u}$ can be $\boldsymbol{u}_D$ (Eq. 5) or $\boldsymbol{u}_C$ (Eq. 6) depending on the fitting method, goodness-of-approx$(\cdot, \cdot)$ is also a function depends on the choice of the fitting method, measuring how likely $\boldsymbol{u}$ is a sample trajectory of $X^{1;\beta_y}$. In Section 3.3, we will define this function alongside the fitting methods.

**Regularizing PETs with latent dynamics.** After learning $g_\gamma$ with Eq. 7, we freeze $\gamma$ and use the goodness-of-approx function as the running cost in Eq. 2 for PETs on downstream tasks. The final objective for PETs becomes

$$\mathcal{L} = \mathcal{L}_T(\boldsymbol{h}_o^{(L)}, y) + \alpha \cdot \text{goodness-of-approx}\big(\boldsymbol{u}(g_\gamma, \{\boldsymbol{h}_o^{(\cdot)}\}), X^{1;\beta_y}\big), \tag{8}$$

where the second term is the running cost and $\alpha$ is a hyper-parameter controlling the regularization strength. By optimizing Eq. 8, PETs need to ensure the final hidden state is capable of predicting $y$, while keeping the latent states at the position $o$ conform to the diffusion bridge $X^{1;\beta_y}$. Note that introducing $g_\gamma$ as the regularizer does not increase the number of trainable parameters for PETs during the training stage since $\gamma$ is fixed. Also, $g_\gamma$ does not involve in the inference stage.

## 3.2 Determining Endpoints for Diffusion Bridges

An intuitive approach to determine the endpoints for the diffusion bridges for each target token is to optimize the endpoints together with the mapping $g_\gamma$. However, optimizing endpoints and $g_\gamma$ jointly may admit a trivial solution: endpoints are both $\boldsymbol{0} \in \mathbb{R}^r$ and $g_\gamma$ always outputs $\boldsymbol{0}$. Since $\boldsymbol{0}$ is always a point in the sample path of such a degenerated diffusion bridge, the value of goodness-of-approximation function can be meaninglessly high. Although sophisticated constraints can be imposed here, as the first work that uses diffusion bridges as regularizers, we simply pre-determine the endpoints and keep them fixed, and remain introducing constraints as our future work.

Specifically, we apply principal component analysis (PCA) to the output token embedding matrix $\boldsymbol{V} \in \mathbb{R}^{|\mathbb{V}|\times d}$ of the PLM, obtaining a $r$-dimensional embedding matrix, and re-normalize each row to have a norm $\eta$. Let the resulting embedding matrix be $\boldsymbol{\beta} \in \mathbb{R}^{|\mathbb{V}|\times r}$. We then use $\boldsymbol{0} \in \mathbb{R}^r$ as the heads for all the bridges, and $\boldsymbol{\beta}$ as the tails of the diffusion bridges, i.e., the $r$-dimensional embedding of $y$ in $\boldsymbol{\beta}$ is used as $\beta_y$ in $X^{1;\beta_y}$. The intuition for using $\boldsymbol{\beta}$ as the tails is that the trajectories of the intermediate states for similar target tokens should be close. In $\boldsymbol{V}$, similar tokens are close, and $\boldsymbol{\beta}$ obtained by PCA can well preserve the token similarity after reducing dimensions.

## 3.3 FITTING THE MAPPING $g_\gamma$

We use the Brownian bridge to illustrate the fitting of $g_\gamma$. It can be analogous to OU bridge easily.

**Method 1: Approximating the Transition PDF.** Generalizing Eq. 4 to high dimension, we can derive the transition PDF from $(0, \mathbf{0})$ to $(t_{i+1}, g_\gamma(\boldsymbol{h}_o^{(i)}, \bar{\boldsymbol{h}}^{(i)}))$ for $X^{1;\beta_y}$:

$$p^{1;\beta_y}(t_{i+1}, g_\gamma(\boldsymbol{h}_o^{(i)}, \bar{\boldsymbol{h}}^{(i)}) \mid 0, \mathbf{0}) \propto \exp(\tfrac{\|g_\gamma(\boldsymbol{h}_o^{(i)}, \bar{\boldsymbol{h}}^{(i)}) - t_{i+1}\beta_y\|^2}{2t_{i+1}(1-t_{i+1})}), \quad (i = 0, \dots, L), \tag{9}$$

where $t_i$ has the same definition as that in $\boldsymbol{u}_D$ (Eq. 5). To make $g_\gamma$ approximate the transition PDF, we maximize the sum of log-probability of $\boldsymbol{u}_D$ under the Brownian bridge $X^{1;\beta_y}$:

$$\text{goodness-of-approx} = \sum_{i=0}^{L} \log\left[p^{1;\beta_y}(t_{i+1}, g_\gamma(\boldsymbol{h}_o^{(i)}, \bar{\boldsymbol{h}}^{(i)}) \mid 0, \mathbf{0})\right] + \text{const.} \tag{10}$$

Here, $g_\gamma$ can be seen as a mapping from the hidden state space to the latent space by predicting the expectation of the Brownian bridge $X^{1;\beta_y}$ at $\{t_{i+1}\}_{i=0}^{L}$.

**Method 2: Approximating the SDE.** Since the Brownian bridge is the solution to the SDE in Eq. 3, we can let $g_\gamma$ approximate the SDE. Solving the SDE requires continuous latent states, while we only have $L+1$ discrete observations, we thus use the continuous representation $\boldsymbol{u}_C$ introduced in Eq. 6. Generalizing Eq. 3 to high dimension, the SDE approximated by $g_\gamma$ can be defined as:

$$dZ_t = g_\gamma(\boldsymbol{h}_o^{(x)}, \bar{\boldsymbol{h}}^{(x)}, t)dt + dB_t, \quad x = (L+2)t - 1, \tag{11}$$

where $x$ is the same as that in Eq. 6, $\boldsymbol{B} : [0, 1] \to \mathbb{R}^r$ is a standard $r$-dimensional Brownian motion. Here, we additionally introduce the dependency on $t$ for $g_\gamma$, since time information is shown to be important in previous neural differential equation works (Zhang et al., 2020; Dupont et al., 2019). Following Li et al. (2020), when two SDEs share the same diffusion function, the KL divergence between the probability measures induced by the two SDEs is finite. Since the diffusion function $\sigma \equiv \boldsymbol{I}$ for Eq. 11 and the multi-dimensional generalization of Eq. 3, the KL divergence between the probability measures $\mu_Y$ of Eq. 11 and $\mu_X$ of generalized Eq. 3 can be estimated by:

$$D_{\text{KL}}(\mu_X || \mu_Y) = \mathbb{E}_Z\left[\int_0^T \frac{1}{2}\|u(t, \gamma)\|_2^2\right],$$
$$u(t, \gamma) = \sigma^{-1}\left(g_\gamma(\boldsymbol{h}_o^{(x)}, \bar{\boldsymbol{h}}^{(x)}, t) - \mu(t, Z_t)\right) = g_\gamma(\boldsymbol{h}_o^{(x)}, \bar{\boldsymbol{h}}^{(x)}, t) - \frac{\beta_y - Z_t}{1 - t}, \tag{12}$$

where $\mu(\cdot, \cdot)$ is the drift function of the pre-determined Brownian bridge $X^{1;\beta_y}$. We use the KL divergence as the goodness-of-approximation function to optimize the mapping $g_\gamma$. Here, $g_\gamma$ can be seen as a mapping from the hidden state space to the latent state space by approximating the drift vector field of the underlying Brownian bridge $X^{1;\beta_y}$.

## 4 EXPERIMENTS

To verify the effectiveness and generality of the regularizers built on stochastic bridges, we conduct experiments on (1) different PLMs: BERT$_{\text{large}}$ (340M) (Devlin et al., 2019) and Deberta$_{\text{xlarge}}$ (750M) (He et al., 2021); (2) different PETs: Prompt tuning, LoRA, BitFit and Adapter; (3) different diffusion bridges: Brownian bridge and OU bridge. We show that the regularizers effectively improve the performance on GLUE (Wang et al., 2019) under both full-set and few-shot settings.

### 4.1 EXPERIMENTAL SETUPS

**Datasets.** Since both BERT$_{\text{large}}$ and Deberta$_{\text{xlarge}}$ use Wikipedia and BookCorpus (Zhu et al., 2015) for pre-training, we thus use these two corpora to train $g_\gamma$. We report accuracy for MNLI, SST-2, QNLI, and RTE; F1 for MRPC and QQP; and Matthews correlation for CoLA. We report the average performance and the standard deviation on the development set over 3 different runs. We append a special token [MASK] to each sequence, and require the PLM to output the label word at [MASK] (e.g., *negative* or *positive* for SST-2). We exclude STS-B for it is a regression task.

**Models and PETs.** We use the checkpoint released by Megatron (Shoeybi et al., 2019) for BERT$_{\text{large}}$, and the official v1 checkpoint for Deberta$_{\text{xlarge}}$. We use a simple three-layer MLP to build $g_\gamma$. For Prompt tuning, we use a soft prompt of length 20, and append it to the end of each

Table 1: The results on GLUE for $BERT_{large}$. The values are the average value of the best performances over three different runs, and the subscripts are the standard deviations. The $\Delta$ column shows the difference of the average performance between the vanilla PETs regularized PETs.

| PET | MNLI | QQP | QNLI | SST-2 | MRPC | CoLA | RTE | Average | $\Delta$ |
|---|---|---|---|---|---|---|---|---|---|
| PROMPT | $84.4_{0.1}$ | $85.3_{0.3}$ | $91.5_{0.1}$ | $95.5_{0.1}$ | $73.9_{2.4}$ | $55.5_{3.4}$ | $60.8_{1.5}$ | $78.1_{0.6}$ | - |
| +BROWN_PDF | $84.7_{0.2}$ | $\mathbf{85.5}_{0.0}$ | $\mathbf{91.8}_{0.6}$ | $95.7_{0.1}$ | $75.4_{0.5}$ | $56.4_{3.3}$ | $61.5_{2.2}$ | $78.7_{0.4}$ | 0.6 |
| +BROWN_SDE | $\mathbf{84.9}_{0.2}$ | $85.4_{0.1}$ | $\mathbf{91.8}_{0.4}$ | $\mathbf{95.8}_{0.3}$ | $\mathbf{78.8}_{1.2}$ | $\mathbf{61.4}_{2.9}$ | $\mathbf{64.7}_{1.1}$ | $\mathbf{80.4}_{0.2}$ | 2.3 |
| LoRA | $88.8_{0.1}$ | $89.2_{0.2}$ | $93.5_{0.2}$ | $95.5_{0.1}$ | $84.6_{0.4}$ | $62.8_{1.6}$ | $78.9_{1.6}$ | $84.8_{0.3}$ | - |
| +BROWN_PDF | $\mathbf{88.9}_{0.1}$ | $\mathbf{89.6}_{0.1}$ | $\mathbf{93.9}_{0.1}$ | $95.6_{0.2}$ | $85.1_{0.7}$ | $63.7_{0.5}$ | $80.0_{0.5}$ | $85.2_{0.1}$ | 0.4 |
| +BROWN_SDE | $\mathbf{88.9}_{0.1}$ | $89.5_{0.1}$ | $93.7_{0.1}$ | $\mathbf{95.7}_{0.1}$ | $\mathbf{86.5}_{1.2}$ | $63.9_{0.4}$ | $\mathbf{80.9}_{0.8}$ | $\mathbf{85.6}_{0.1}$ | 0.8 |
| BitFit | $87.9_{0.2}$ | $87.6_{0.1}$ | $92.7_{0.2}$ | $95.6_{0.1}$ | $79.4_{2.3}$ | $60.2_{0.8}$ | $77.0_{1.5}$ | $82.9_{0.3}$ | - |
| +BROWN_PDF | $87.9_{0.1}$ | $\mathbf{87.8}_{0.0}$ | $\mathbf{93.0}_{0.2}$ | $95.7_{0.1}$ | $83.1_{0.8}$ | $60.3_{0.6}$ | $\mathbf{78.3}_{0.9}$ | $83.7_{0.2}$ | 0.8 |
| +BROWN_SDE | $87.9_{0.2}$ | $87.7_{0.0}$ | $92.8_{0.1}$ | $95.7_{0.1}$ | $\mathbf{83.3}_{0.8}$ | $\mathbf{61.1}_{1.2}$ | $77.7_{1.5}$ | $\mathbf{83.8}_{0.3}$ | 0.9 |
| Adapter | $88.8_{0.1}$ | $89.6_{0.3}$ | $93.7_{0.1}$ | $95.6_{0.1}$ | $83.6_{0.1}$ | $60.4_{1.2}$ | $79.5_{1.2}$ | $84.5_{0.3}$ | - |
| +BROWN_PDF | $\mathbf{89.0}_{0.1}$ | $89.7_{0.2}$ | $93.8_{0.2}$ | $\mathbf{95.8}_{0.1}$ | $\mathbf{86.5}_{1.1}$ | $\mathbf{62.6}_{0.7}$ | $\mathbf{83.2}_{0.2}$ | $\mathbf{85.8}_{0.2}$ | 1.3 |
| +BROWN_SDE | $88.9_{0.1}$ | $\mathbf{89.8}_{0.1}$ | $\mathbf{93.9}_{0.2}$ | $\mathbf{95.8}_{0.1}$ | $85.9_{0.4}$ | $62.3_{1.8}$ | $82.2_{0.2}$ | $85.5_{0.2}$ | 1.0 |

sequence. For LoRA, we apply it to the query and value of attention modules. For Adapter, we apply it to the output of attention and feed-forward modules. For BitFit, we tune all the bias terms in linear layers and layer normalization modules. Hereafter, we use **PDF regularizer** to refer to using $g_\gamma$ fitted by approximating the transition PDF, and **SDE regularizer** to refer to using $g_\gamma$ fitted by approximating the SDE, **vanilla** $x$ to refer to the PET $x$ without using regularizers.

**Few-shot Experiments.** We randomly sample $2 \times k$ examples from the original training set $\mathcal{D}_{train}$ for each class. The sampling is performed 5 times with different seeds to form 5 training sets and development sets $\{\tilde{\mathcal{D}}_{train}^{(i)}, \tilde{\mathcal{D}}_{dev}^{(i)}\}_{i=1}^5$ with each being $k$-shot. Each time we train PETs on $\tilde{\mathcal{D}}_{train}^{(i)}$, we select the best model on $\tilde{\mathcal{D}}_{dev}^{(i)}$, and report its performance on the original development set $\mathcal{D}_{dev}$.

**Hyper-parameters.** We list hyper-parameters in Appendix E. We mainly focus on the difference in performance between vanilla PETs and regularized PETs. Therefore, we directly set the hyper-parameters to reasonable values and do not perform much hyper-parameter search. But we ensure the hyper-parameters for vanilla PETs and regularized PETs are the same for a fair comparison.

## 4.2 FULL-SET RESULTS

The experimental results for $BERT_{large}$ are reported in Table 1. Due to space limitation, see Table 7 for the complete results including OU bridge regularizers. The first line of each block in the table is the performance of vanilla PETs, and the rest of the lines are the performances of the regularized PETs. We also conduct the same experiments for $Deberta_{xlarge}$ and place the results in Appendix C.

In general, both Brownian and OU bridges, and both PDF and SDE regularizers are able to effectively improve the performance of PETs, showing the effectiveness of our proposed regularizers. Particularly, for Prompt tuning, the SDE regularizer with both diffusion bridges yield an increase on the average performance of more than 2 points. We assume that it is because Prompt tuning has far less trainable parameters than other PETs, and it only acts at the input layer, which is far from the supervision signals of the terminal cost $\mathcal{L}_T$. Therefore, when provided with the regularization on the hidden states, the prompts receive more guidance and eventually reaching a better local optimal.

Overall, the two diffusion bridges in our experiments do not show much difference. As for the two fitting methods, SDE regularizer is generally more effective, especially for Prompt tuning where the number of trainable parameters is restricted. However, we also observe that SDE regularizer is about 3 times slower than PDF regularizer, which brings the trade-off between performance and efficiency. One can expect a better performance by leveraging more sophisticated underlying stochastic bridges, exploring more reasonable endpoints for bridges and designing better mapping $g_\gamma$. As the first work using latent stochastic bridges as regularizers, we mainly consider the most straightforward cases and aim to show the great potential of the approach and the importance of regularizing hidden states.

## 4.3 FEW-SHOT RESULTS

We observe that in Table 1, the improvements brought by our regularizers are more substantial on small datasets MRPC, CoLA and RTE. We assume that this is because in rich-resource datasets,

Table 2: The results on GLUE for BERT$_{\text{large}}$ under the 16-shot setting. We exclude CoLA because all PETs fail to give reasonable results under the few-shot setting.

| PET | MNLI | QQP | QNLI | SST-2 | MRPC | RTE | Average | $\Delta$ |
|---|---|---|---|---|---|---|---|---|
| PROMPT | $38.1_{1.5}$ | $53.0_{3.1}$ | $51.6_{1.4}$ | $70.1_{4.9}$ | $50.1_{3.0}$ | $48.0_{1.3}$ | $51.8_{0.9}$ | - |
| +BROWN_PDF | $38.7_{2.3}$ | $54.9_{2.8}$ | $52.1_{1.1}$ | $75.0_{11.0}$ | $\mathbf{52.8}_{2.2}$ | $50.8_{3.3}$ | $54.0_{1.8}$ | 2.2 |
| +BROWN_SDE | $\mathbf{40.6}_{0.8}$ | $\mathbf{55.4}_{2.1}$ | $\mathbf{52.9}_{1.6}$ | $\mathbf{80.0}_{10.9}$ | $51.9_{3.6}$ | $\mathbf{51.7}_{3.1}$ | $\mathbf{55.4}_{1.5}$ | $\mathbf{3.6}$ |
| LORA | $48.7_{4.5}$ | $59.9_{5.5}$ | $53.2_{1.2}$ | $90.2_{1.1}$ | $53.6_{3.4}$ | $64.2_{0.9}$ | $61.6_{0.6}$ | - |
| +BROWN_PDF | $52.0_{1.3}$ | $62.7_{2.2}$ | $55.1_{3.8}$ | $\mathbf{91.3}_{0.1}$ | $57.4_{5.0}$ | $65.3_{1.5}$ | $64.0_{0.7}$ | 2.4 |
| +BROWN_SDE | $\mathbf{54.1}_{0.9}$ | $\mathbf{65.5}_{1.4}$ | $\mathbf{64.3}_{5.1}$ | $91.2_{0.3}$ | $\mathbf{60.2}_{3.1}$ | $\mathbf{65.8}_{1.4}$ | $\mathbf{66.8}_{0.6}$ | $\mathbf{5.2}$ |
| BITFIT | $48.4_{1.6}$ | $56.0_{6.1}$ | $51.7_{2.5}$ | $90.8_{0.8}$ | $52.0_{2.3}$ | $61.7_{1.4}$ | $60.1_{1.1}$ | - |
| +BROWN_PDF | $48.5_{1.9}$ | $56.0_{6.0}$ | $53.5_{2.0}$ | $\mathbf{91.0}_{0.4}$ | $53.9_{2.3}$ | $63.2_{1.6}$ | $61.0_{0.8}$ | 0.9 |
| +BROWN_SDE | $\mathbf{52.3}_{0.5}$ | $\mathbf{61.2}_{2.9}$ | $\mathbf{58.8}_{4.7}$ | $90.8_{0.4}$ | $\mathbf{54.8}_{3.0}$ | $\mathbf{63.9}_{2.5}$ | $\mathbf{63.6}_{0.8}$ | $\mathbf{3.5}$ |
| ADAPTER | $47.4_{3.7}$ | $57.0_{7.2}$ | $55.8_{2.9}$ | $91.0_{0.4}$ | $55.8_{2.5}$ | $62.7_{2.0}$ | $61.6_{1.2}$ | - |
| +BROWN_PDF | $49.0_{4.8}$ | $58.5_{7.4}$ | $56.9_{3.1}$ | $91.4_{0.2}$ | $57.2_{4.9}$ | $63.2_{3.0}$ | $62.7_{1.5}$ | 1.1 |
| +BROWN_SDE | $\mathbf{52.3}_{2.2}$ | $\mathbf{62.4}_{2.9}$ | $\mathbf{64.8}_{4.5}$ | $\mathbf{91.9}_{0.4}$ | $\mathbf{57.3}_{4.1}$ | $\mathbf{63.8}_{1.8}$ | $\mathbf{65.4}_{1.3}$ | $\mathbf{3.8}$ |

the abundant data itself has provided enough information to learn high quality PETs, while in low-resource datasets, the data is insufficient and the regularizer can offer additional helpful supervision. To validate this, we conduct the experiments under the few-shot setting on GLUE.

The 16-shot results are shown in Table 2, and the results for the OU bridge, results for 4-, 8- and 32-shot and results for Deberta$_{\text{xlarge}}$ are placed in Appendix C. For all PETs, the SDE regularizer yields an improvement of more than 3 points. Particularly, the SDE regularizer on LoRA brings an improvement of 5.2 points. By applying regularizers under the few-shot setting, there is now a substantial boost on what was originally a rich-resource dataset, such as MNLI, QQP and QNLI. The PDF regularizer also gives modest improvements. Although slightly inferior to the SDE regularizer, it is still satisfying, considering that the PDF regularizer brings such a performance improvement with little computational cost introduced. We additionally observe that the improvement is more significant on Deberta$_{\text{xlarge}}$ in Table 9, demonstrating the potential of our regularizers on larger models.

## 5 ANALYSES

In order to have a better understanding of the role played by our regularizers, we analyze the hidden states of the PETs with and without regularizers in this section. We choose Prompt tuning as a representative in the analyses. By varying the hyper-parameter $\alpha$ in Eq. 8, we show that as the strength of the regularization gets stronger, the clusters of hidden states corresponding to different label tokens become compacter and more distinguishable. Also, we show that the hidden states of vanilla PETs spontaneously approach the latent bridges in the latent space without knowing the bridges, indicating that there may exist intrinsically diffusion-bridge-like latent dynamics for PETs.

### 5.1 THE REGULARIZER WIDENS THE DISTANCES BETWEEN LABEL CLUSTERS

We use the different prompts obtained with or without regularizers on the full-set GLUE, and record the intermediate hidden states at the position of prediction as $\{h_{[\text{MASK}]}^{(i)}\}_{i=1}^{L}$. We vary the regularization strength by adjusting the coefficient $\alpha$ in Eq. 8 to inspect the impact of the regularization strength on the hidden states. Note that when $\alpha = 0$, it degenerates to the vanilla PET.

We randomly sample 100 samples for each label in MNLI, and use UMAP (McInnes et al., 2018) to reduce the dimension of the last layer's hidden states of different prompts, and plot them in Figure 2. It shows clearly that for both regularizers, as the regularization strength becomes stronger, the hidden states of the last layer becomes more distinguishable among different labels. By looking at the axes of these plots, we find that the distance between the clusters generally increases when the reg-

Table 3: Pearson's correlation between the regularization strength $\alpha$ and the average distance between the centroids of different label clusters. ***: $p < .001$, **: $p < .01$, *: $p < .05$.

| | Coef. | |
|---|---|---|
| Dataset | PDF | SDE |
| MNLI | 0.928*** | 0.894*** |
| QQP | 0.715* | 0.897*** |
| QNLI | 0.971*** | 0.837** |
| SST-2 | 0.966*** | 0.867*** |
| MRPC | 0.226 | 0.433 |
| CoLA | 0.633* | 0.807** |
| RTE | 0.440 | 0.589* |

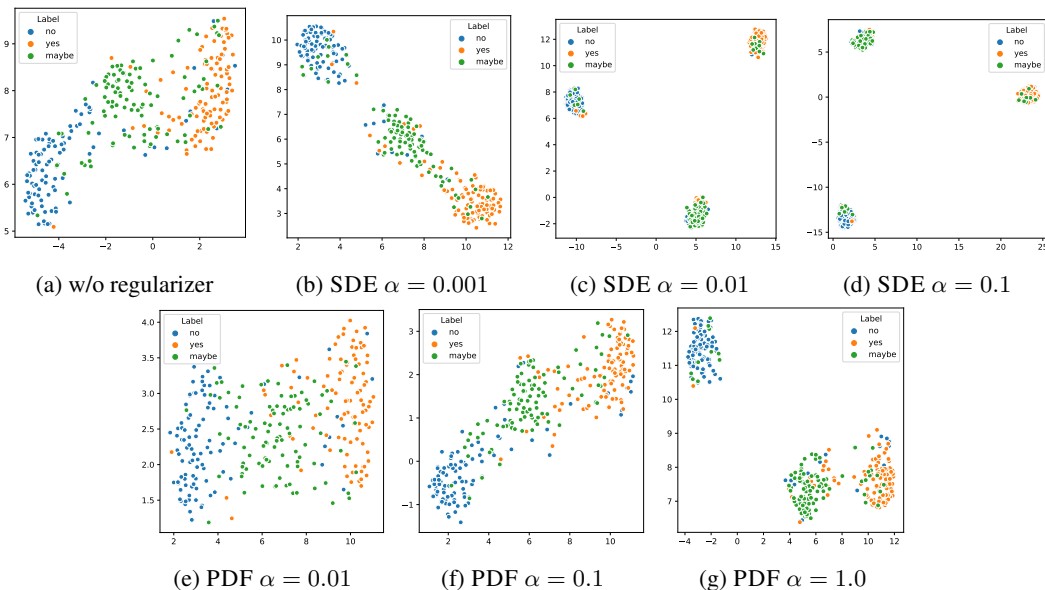

(a) w/o regularizer    (b) SDE $\alpha = 0.001$    (c) SDE $\alpha = 0.01$    (d) SDE $\alpha = 0.1$

(e) PDF $\alpha = 0.01$    (f) PDF $\alpha = 0.1$    (g) PDF $\alpha = 1.0$

Figure 2: A visualization of the last layer's hidden states on MNLI using the prompt that is trained (a) without regularizer (b-d) with the SDE regularizer (e-g) with the PDF regularizer.

ularization strength is increased. We also notice that the SDE regularizer better helps separate the hidden states of the last layer by substantially enlarging the distance between the centroids of different labels, which could be one of the reasons why the SDE regularizer has a better effectiveness in almost all our experiments.

We also calculate the Pearson's correlation between the regularization strength $\alpha$ and the average distance between the centroids of different clusters. The results are shown in Table 3. On all the datasets, the regularization strength $\alpha$ has a positive correlation to the average centroid distance, and on most of the datasets, the correlations are significant in the sense that the $p$-value $< .05$. This indicates that as the regularization strength becomes stronger, the centroids of different label clusters become more distant, which is a desired effect because the regularizer encourages the hidden states for different label tokens to conform to different latent diffusion bridges.

## 5.2   HIDDEN STATES APPROACH SPONTANEOUSLY TO THE LATENT BRIDGES

An interesting phenomenon we observe is that the vanilla PETs' intermediate hidden states spontaneously approach our latent bridges when they are projected by our mapping $g_\gamma$. That is, applying our mapping $g_\gamma$ to the hidden states of vanilla PETs, we find that when the performance of vanilla PETs becomes better, the average distance from $g_\gamma(\{\boldsymbol{h}_o^{(\cdot)}, \bar{\boldsymbol{h}}^{(\cdot)}\})$ to our latent bridge gets closer. Here, similar to Wang et al. (2022), we define the distance from $g_\gamma(\boldsymbol{h}_o^{(\cdot)}, \bar{\boldsymbol{h}}^{(\cdot)})$ to its corresponding latent bridge $X_y$ using Eq. 10 without the constant. Note that the vanilla PETs have no access to $g_\gamma$ and the latent bridges during the training process, and $g_\gamma$ also has no access to the PETs during its fitting process.

Table 4: Kendall's rank correlation between the number of shots and the distance to the latent bridges.

| Dataset | Coef. | $p$-value |
|---------|-------|-----------|
| MNLI    | -0.39 | .026*     |
| QQP     | -0.36 | .037*     |
| QNLI    | -0.32 | .069      |
| SST-2   | -0.50 | .005**    |
| MRPC    | -0.21 | .225      |
| RTE     | -0.34 | .049*     |

We demonstrate the above observation by conducting analyses in few-shot scenarios and reporting the correlation between (1) the number of shots and the average distance from latent hidden states to latent bridges in Table 4 (2) the performance and the average distance from latent hidden states to latent bridges in Table 5. Specifically, we report Kendall's rank correlation[2] for (1), and Pearson's correlation for (2). See Appendix D for the detailed setup for the calculation of the correlations.

---

[2]The reason that we choose Kendall's rank correlation is that it is suitable for data with ties. See Appendix D.

From Table 4, the number of shots has a negative correlation to the distance, and the correlation is significant on 4 out of 6 datasets. This indicates that as the amount of available data increases for vanilla PETs, its intermediate hidden states in latent space spontaneously approach latent bridges even without knowing the mapping $g_\gamma$ and the bridges. Additionally, the results in Table 5 show the negative correlation between the performance of vanilla PETs and the distance to the latent bridges, and it is significant on 3 out of 6 datasets.

Altogether, the two findings on correlation show that as the PETs learn to solve the problem, their intermediate hidden states spontaneously approach our latent bridges in the latent space projected by $g_\gamma$. This implies that there exists intrinsically diffusion-bridge-like latent dynamics for PETs, and also justifies our use of latent diffusion bridges as regularizers.

Table 5: Pearson's correlation between the performance and the distance to the latent bridges.

| Dataset | Coef. | $p$-value |
|---------|-------|-----------|
| MNLI | -0.09 | .715 |
| QQP | -0.42 | .062 |
| QNLI | -0.77 | $<.001^{***}$ |
| SST-2 | -0.69 | $<.001^{***}$ |
| MRPC | -0.13 | .591 |
| RTE | -0.87 | $<.001^{***}$ |

## 6 RELATED WORKS

Recent years have witnessed the success of PLMs (Raffel et al., 2020; Brown et al., 2020), which acquires rich knowledge from unlabeled data in a self-supervised manner and can be adapted to specific tasks by tuning their parameters. Despite the success of PLMs, as the size of PLMs continues to grow, it becomes increasingly impractical to perform full-parameter fine-tuning for downstream tasks. Therefore, various recent efforts have been devoted to PETs, aiming to freeze PLMs and only tune a few additional parameters for task adaptation, such as Prompt tuning (Lester et al., 2021) that prepends tunable tokens per task to the input text, Adapter (Houlsby et al., 2019) that inserts small modules into each layer, BitFit (Zaken et al., 2022) that tunes only the bias terms, and LoRA (Hu et al., 2022) that decomposes the weight updates into low-rank matrices. In this paper, based on the theoretical grounding of PETs on optimal control (Yang & Liu, 2022; Ding et al., 2022), we develop stochastic bridges as the regularizer for intermediate hidden states and introduce regularized PETs, showing effectiveness and generality on different PLMs and tasks.

Since we adopt latent stochastic bridges as the regularizer, our work closely relates to continuous-time neural ODEs (Chen et al., 2018; Rubanova et al., 2019) and neural SDEs (Li et al., 2019; Kidger et al., 2021). Continuous-time neural ODEs and SDEs model the dynamics of the hidden states with ODEs and SDEs parameterized by neural network respectively For example Chen et al. (2018) show the resemblance between ODE and ResNet (He et al., 2016), and propose to learn the neural ODEs as the generalization of ResNet in continuous time. Inspired by these works, we use SDEs to represent the latent dynamics of PETs in the latent space. Our work differs from these work in that we focus on using neural SDEs as regularizers for intermediate hidden states, rather than feature extractors on downstream tasks.

We also notice that Wang et al. (2022) explore the use of Brownian bridge in PLMs. However, they use Brownian bridge to regularize the text dynamics across time, while we use Brownian bridge to regularize the dynamics of intermediate hidden states across model's layers. We additionally show that diffusion bridges other than Brownian bridge can be easily applied in our regularizer. As far as we know, our work is the first to show the diffusion-bridge-like dynamics for hidden states across layers and use diffusion bridges as the regularizer for intermediate hidden states.

## 7 CONCLUSION

In this work, we start from the optimal control perspective of PETs and we notice that the existing PETs lack a running cost that regularizes the intermediate hidden states. Therefore, we propose to use stochastic bridges in a latent space as the regularizers for PETs. Experimental results on different models, different tasks and different PETs show that the proposed regularizer effectively improve the PETs' performance. Our analyses further show that when the PETs are trained without the regularizer, the hidden states spontaneously approach our diffusion bridges, indicating that there exists intrinsically diffusion-bridge-like dynamics for PETs. As the first work using stochastic bridges as regularizers, we show its effectiveness and generality even with simple diffusion bridges. We believe it is a promising direction and we are excited to see more future work.

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

## A  BACKGROUND FOR PARAMETER-EFFICIENT TUNING METHODS

The large number of parameters in the PLMs makes fine-tuning impractical, therefore different PETs are proposed to mitigate the problem. The current PETs can be categorized into three groups: addition-based, specification-based and reparameterization-based (Ding et al., 2022). To verify the generality of our method, we include one or two PETs from each category in this work, and we give a brief review to these PETs.

**Prompt Tuning** is an addition-based PET. It prepends or appends trainable virtual tokens $\boldsymbol{P} \in \mathbb{R}^{m \times d}$ to each sequence $\boldsymbol{x} \in \mathbb{R}^{n \times d}$ to form a new input sequence $[\boldsymbol{P}; \boldsymbol{x}]$ or $[\boldsymbol{x}; \boldsymbol{P}] \in \mathbb{R}^{(n+m) \times d}$, where $n, m$ are length of original sequence and virtual tokens respectively, $d$ is the embedding dimension. The virtual tokens $\boldsymbol{P}$ can be either continuous Lester et al. (2021) or be restricted to be embeddings of discrete tokens in vocabulary (Gao et al., 2021).

**Adapter** (Houlsby et al., 2019) is an addition-based PET. It inserts two-layer MLPs after the attention module and feed-forward module at each layer. Denote $\boldsymbol{h} \in \mathbb{R}^d$ as the input of Adapter, $r$ as the intermediate dimension of Adapter's MLP, $\boldsymbol{W}_d \in \mathbb{R}^{r \times d}, \boldsymbol{W}_u \in \mathbb{R}^{d \times r}$ as the down-projection and up-projection of Adapter, and $\sigma$ as the activation function. Then the computation of Adapter can be formulated as

$$\boldsymbol{h} \leftarrow \boldsymbol{W}_u \sigma(\boldsymbol{W}_d \boldsymbol{x}) + \boldsymbol{h}$$

**BitFit** (Zaken et al., 2022) is a specification-based PET. It specifies the bias terms in layer normalization modules and linear transformation modules as trainable.

**LoRA** (Hu et al., 2022) is a reparameterization-based PET. It assumes that when training the model, the updates $\Delta \boldsymbol{W}$ for model's pre-trained parameters $\boldsymbol{W} \in \mathbb{R}^{d \times k}$ are low-rank, and thus reparameterize the $\Delta \boldsymbol{W}$ of each matrix in attention module with a low-rank decomposition $\Delta \boldsymbol{W} = \boldsymbol{B} \boldsymbol{A}$, where $\boldsymbol{B} \in \mathbb{R}^{d \times r}, \boldsymbol{A} \in \mathbb{R}^{r \times k}$. For a forward pass $\boldsymbol{h} = \boldsymbol{W} \boldsymbol{x}$, the computation of LoRA can be written as

$$\boldsymbol{h} = (\boldsymbol{W} + \Delta \boldsymbol{W}) \boldsymbol{x} = \boldsymbol{W} \boldsymbol{x} + \boldsymbol{B} \boldsymbol{A} \boldsymbol{x}$$

## B  PROPERTIES FOR ORNSTEIN-UHLENBECK BRIDGE

**Proposition B.1** (Properties of Ornstein-Uhlenbeck Bridge). A Ornstein-Uhlenbeck $X^{T;\beta}$ pinned at $X_0^{T;\beta} = 0$ and $X_T^{T;\beta} = \beta$ is the solution to the following SDE:

$$d\tilde{X}_t = q \left[ -\coth\left[q(T-t)\right] \tilde{X}_t + \frac{\beta}{\sinh\left[q(T-t)\right]} \right] dt + \sigma dB_t, \quad \tilde{X}_0 = 0, \tag{13}$$

where $q$ is the diffusion coefficient and $\sigma$ is the diffusion for the OU process. The transition probability density function reads as:

$$p^{T;\beta}(t, y \mid 0, 0) = \frac{1}{\sqrt{2\pi\sigma(s,t)}} \exp\left\{ -\frac{\left( y - \frac{\sinh(q(T-t))}{\sinh(q(T-s))}\beta \right)^2}{2\sigma(s,t)} \right\}, \tag{14}$$

where

$$\sigma(s,t) = \frac{\sigma^2}{q} \frac{\sinh(q(T-t))\sinh(q(t-s))}{\sinh(q(T-s))}. \tag{15}$$

## C  OTHER RESULTS FOR GLUE EXPERIMENTS

In this section, we present the complete results including OU bridge regularizer for Table 1 and Table 2. We also report the results for Deberta$_{\text{xlarge}}$, and the results on few-shot GLUE for both

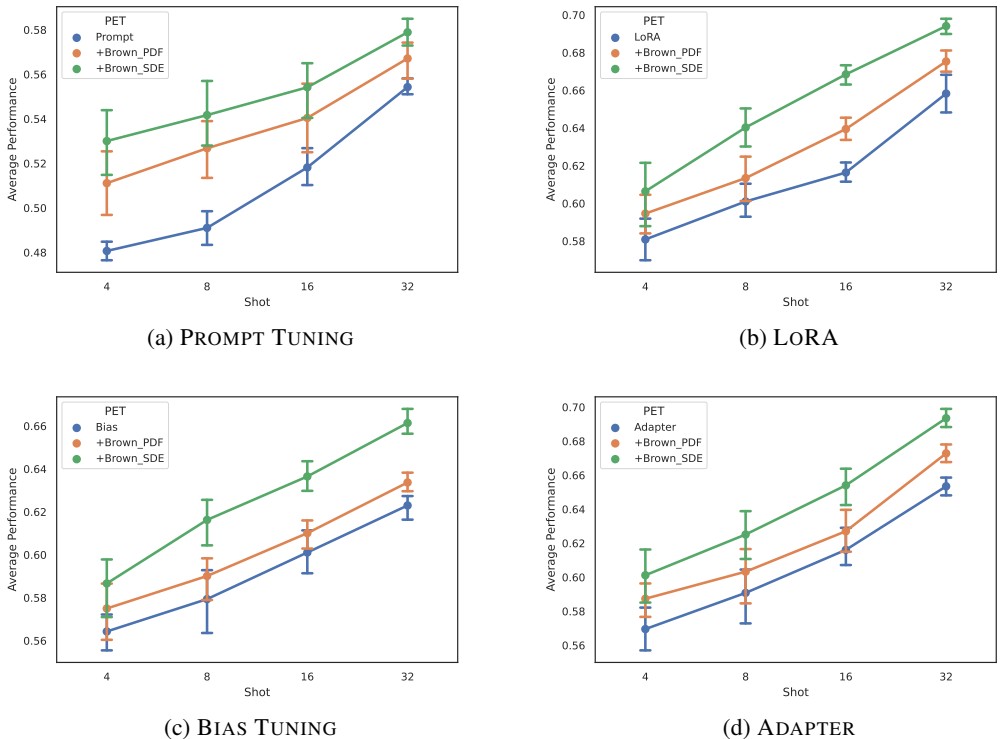

(a) PROMPT TUNING

(b) LoRA

(c) BIAS TUNING

(d) ADAPTER

Figure 3: The average BERT$_{large}$ few-shot GLUE results trained with different PETs under different shots. The results are averaged across 5 different seeds and the error bars indicate the 95% confidence. SDE regularizer consistently outperforms the baseline PDF regularizer.

BERT$_{large}$ and Deberta$_{xlarge}$ under 4-, 8-, 16-, and 32-shot. We observe that BERT$_{large}$ cannot give reasonable answers on CoLA and the Matthews correlations are around 0 for all the PETs and all the shots we have experienced with. However, the situation gets better for the larger model Deberta$_{xlarge}$. Therefore, we exclude CoLA for BERT$_{large}$ and keep it for Deberta$_{xlarge}$. We only select the Brownian bridge as the representative in this section, since the Brownian bridge and Ornstein-Uhlenbeck bridge have no significant difference in Table 1.

In Table 7 and Table 6, we report the performance of OU bridge regularizers. The experimental setups are the same as Table 1 and Table 2 respectively. The performances between OU bridge and Brownian bridge do not have a significant difference. In Table 8, we report the performance of Deberta$_{xlarge}$ on full GLUE datasets. On all four PETs, the SDE regularizer outperforms the PDF regularizer, this is consistent with the results we see in Table 1. The results for 4-, 8-, and 32-shot for BERT$_{large}$ and Deberta$_{xlarge}$ are plotted respectively in Figure 3 and Figure 4. For simplicity, we only plot the average performance for each PET. We report the results for 16-shot experiments for Deberta$_{xlarge}$ in Table 9. The setup for the experiment is almost the same as the experiment in Section 4.3, and the hyper-parameters are listed in Appendix E. The SDE regularizer outperforms the PDF regularizer on most of the PETs except Prompt tuning. We notice that the SDE regularizer helps Deberta$_{xlarge}$ substantially on CoLA for most of the PETs, indicating the SDE regularizer can effectively provide useful guidance when the data is scarce and the task is hard.

## D CALCULATION OF CORRELATION IN SECTION 5

In this section, we elaborate on how we calculate the correlations reported in Table 4 and Table 5.

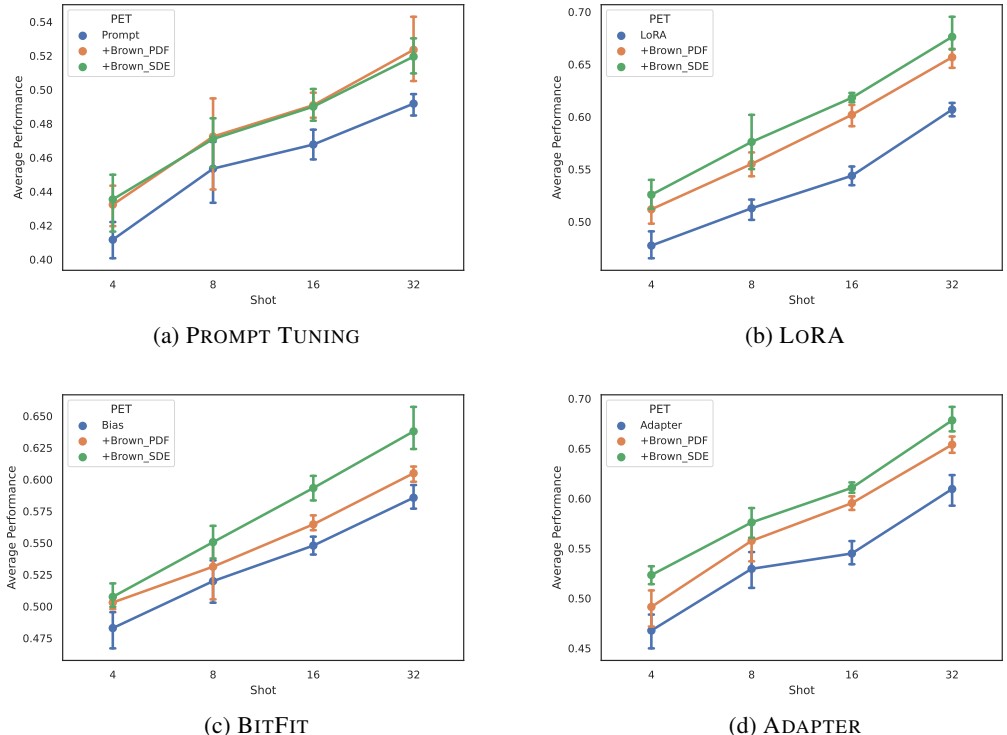

Figure 4: The average Deberta$_{\text{xlarge}}$ few-shot GLUE results trained with different PETs under different shots. The results are averaged across 5 different seeds and the error bars indicate the 95% confidence.

## D.1 CORRELATION BETWEEN NUMBER OF SHOTS AND DISTANCE TO BRIDGE

**Definition D.1** (Tie). A pair of observation $\{(x_i, y_i), (x_j, y_j)\}$ is defined as tied if $x_i = x_j$ or $y_i = y_j$.

Since we generate the few-shot datasets using 5 random seeds for each shot, each PET has 5 results for each shot. This results in observations with ties, e.g., the two observations for distances on the first seed and second seed for 8-shot $\{(8, d_1), (8, d_2)\}$ are tied. To calculate the correlation for data with ties, the Tau-b of Kendall's rank correlation is more suitable than Pearson's correlation. We therefore report the Kendall's rank correlation for the correlation between the number of shots and the hidden states distance to the latent bridges.

## D.2 CORRELATION BETWEEN PERFORMANCE AND DISTANCE TO BRIDGE

We mix all the few-shot results for different shots and different seeds to form observations of performance and the hidden states distances to bridges, and then calculate the Pearson's correlation.

## E HYPER-PARAMETERS

### E.1 TRAINING $g_\gamma$

we use simple 3-layer MLP for $g_\gamma$ in all of our experiments. For PDF regularizer, the output dimensions of each layer are 1024, 256, 128, and for SDE regularizer, the output dimensions of each layer are 1024, 256, 32. We observe that the running time increases noticeably with the final output dimension for SDE regularizer, we thus choose a smaller one for SDE regularizer. The hyper-parameters for training $g_\gamma$ are listed in Table 10.

Table 6: The complete results on GLUE for BERT$_\text{large}$ under 16-shot setting. We exclude CoLA because all PETs fail to give reasonable result in few-shot setting.

| PET | MNLI | QQP | QNLI | SST-2 | MRPC | RTE | Average | Δ |
|---|---|---|---|---|---|---|---|---|
| PROMPT | $38.1_{1.5}$ | $53.0_{3.1}$ | $51.6_{1.4}$ | $70.1_{4.9}$ | $50.1_{3.0}$ | $48.0_{1.3}$ | $51.8_{0.9}$ | - |
| +BROWN_PDF | $38.7_{2.3}$ | $54.9_{2.8}$ | $52.1_{1.1}$ | $75.0_{11.0}$ | $52.8_{2.2}$ | $50.8_{3.3}$ | $54.0_{1.8}$ | 2.2 |
| +BROWN_SDE | $\mathbf{40.6}_{0.8}$ | $55.4_{2.1}$ | $52.9_{1.6}$ | $80.0_{10.9}$ | $51.9_{3.6}$ | $51.7_{3.1}$ | $55.4_{1.5}$ | 3.6 |
| +OU_PDF | $38.9_{1.6}$ | $54.9_{2.5}$ | $51.8_{2.0}$ | $71.4_{9.3}$ | $\mathbf{54.3}_{2.6}$ | $51.5_{3.5}$ | $53.8_{1.9}$ | 2.0 |
| +OU_SDE | $40.0_{1.7}$ | $\mathbf{56.8}_{1.5}$ | $\mathbf{53.5}_{2.2}$ | $\mathbf{82.7}_{3.2}$ | $52.7_{2.1}$ | $\mathbf{52.0}_{1.9}$ | $\mathbf{56.3}_{0.5}$ | $\mathbf{4.5}$ |
| LoRA | $48.7_{4.5}$ | $59.9_{5.5}$ | $53.2_{1.2}$ | $90.2_{1.1}$ | $53.6_{3.4}$ | $64.2_{0.9}$ | $61.6_{0.6}$ | - |
| +BROWN_PDF | $52.0_{1.3}$ | $62.7_{2.2}$ | $55.1_{3.8}$ | $91.3_{0.1}$ | $57.4_{5.0}$ | $65.3_{1.5}$ | $64.0_{0.7}$ | 2.4 |
| +BROWN_SDE | $\mathbf{54.1}_{0.9}$ | $\mathbf{65.5}_{1.4}$ | $64.3_{5.1}$ | $91.2_{0.3}$ | $60.2_{3.1}$ | $65.8_{1.4}$ | $\mathbf{66.8}_{0.6}$ | $\mathbf{5.2}$ |
| +OU_PDF | $51.6_{2.4}$ | $62.5_{2.3}$ | $55.3_{3.4}$ | $\mathbf{91.4}_{0.5}$ | $56.9_{5.2}$ | $65.6_{0.4}$ | $63.9_{1.1}$ | 2.3 |
| +OU_SDE | $52.9_{2.6}$ | $63.1_{1.5}$ | $\mathbf{64.4}_{5.1}$ | $91.3_{0.3}$ | $\mathbf{62.2}_{2.7}$ | $\mathbf{66.2}_{1.1}$ | $66.7_{0.7}$ | 5.1 |
| BITFIT | $48.4_{1.6}$ | $56.0_{6.1}$ | $51.7_{2.5}$ | $90.8_{0.8}$ | $52.0_{2.3}$ | $61.7_{1.4}$ | $60.1_{1.1}$ | - |
| +BROWN_PDF | $48.5_{1.9}$ | $56.0_{6.0}$ | $53.5_{2.0}$ | $91.0_{0.4}$ | $53.9_{2.3}$ | $63.2_{1.6}$ | $61.0_{0.8}$ | 0.9 |
| +BROWN_SDE | $\mathbf{52.3}_{0.5}$ | $\mathbf{61.2}_{2.9}$ | $58.8_{4.7}$ | $90.8_{0.4}$ | $54.8_{3.0}$ | $\mathbf{63.9}_{2.5}$ | $\mathbf{63.6}_{0.8}$ | $\mathbf{3.5}$ |
| +OU_PDF | $49.0_{1.8}$ | $56.1_{6.1}$ | $52.8_{2.2}$ | $90.8_{0.7}$ | $53.5_{2.2}$ | $62.5_{1.3}$ | $60.8_{0.8}$ | 0.7 |
| +OU_SDE | $51.8_{1.2}$ | $58.4_{2.2}$ | $\mathbf{58.9}_{4.1}$ | $\mathbf{91.1}_{0.4}$ | $\mathbf{56.4}_{3.6}$ | $63.5_{1.4}$ | $63.4_{0.8}$ | 3.3 |
| ADAPTER | $47.4_{3.7}$ | $57.0_{7.2}$ | $55.8_{2.9}$ | $91.0_{0.4}$ | $55.8_{2.5}$ | $62.7_{2.0}$ | $61.6_{1.2}$ | - |
| +BROWN_PDF | $49.0_{4.8}$ | $58.5_{7.4}$ | $56.9_{3.1}$ | $91.4_{0.2}$ | $57.2_{4.9}$ | $63.2_{3.0}$ | $62.7_{1.5}$ | 1.1 |
| +BROWN_SDE | $\mathbf{52.3}_{2.2}$ | $62.4_{2.9}$ | $\mathbf{64.8}_{4.5}$ | $\mathbf{91.9}_{0.4}$ | $57.3_{4.1}$ | $\mathbf{63.8}_{1.8}$ | $\mathbf{65.4}_{1.3}$ | $\mathbf{3.8}$ |
| +OU_PDF | $48.4_{4.3}$ | $58.4_{7.4}$ | $57.7_{3.7}$ | $91.3_{0.5}$ | $57.1_{3.3}$ | $63.0_{1.8}$ | $62.6_{1.3}$ | 1.0 |
| +OU_SDE | $48.3_{4.5}$ | $\mathbf{62.8}_{1.8}$ | $63.8_{4.7}$ | $91.1_{0.4}$ | $\mathbf{59.8}_{2.3}$ | $63.6_{1.9}$ | $64.9_{0.6}$ | 3.3 |

Table 7: The complete results on GLUE for BERT$_\text{large}$. The values are the average value of the best performances over three different runs, and the subscripts are the standard deviations. The Δ column shows the difference of the average performance between the PETs with and without our regularizers.

| PET | MNLI | QQP | QNLI | SST-2 | MRPC | CoLA | RTE | Average | Δ |
|---|---|---|---|---|---|---|---|---|---|
| PROMPT | $84.4_{0.1}$ | $85.3_{0.3}$ | $91.5_{0.1}$ | $95.5_{0.1}$ | $73.9_{2.4}$ | $55.5_{3.4}$ | $60.8_{1.5}$ | $78.1_{0.6}$ | - |
| +BROWN_PDF | $84.7_{0.2}$ | $\mathbf{85.5}_{0.0}$ | $91.8_{0.6}$ | $95.7_{0.1}$ | $75.4_{0.5}$ | $56.4_{3.3}$ | $61.5_{2.2}$ | $78.7_{0.4}$ | 0.6 |
| +OU_PDF | $84.7_{0.1}$ | $85.4_{0.1}$ | $91.8_{0.3}$ | $95.6_{0.2}$ | $76.9_{1.0}$ | $57.1_{1.2}$ | $60.5_{3.0}$ | $78.9_{0.2}$ | 0.8 |
| +BROWN_SDE | $\mathbf{84.9}_{0.2}$ | $85.4_{0.1}$ | $91.8_{0.4}$ | $\mathbf{95.8}_{0.3}$ | $78.8_{1.2}$ | $61.4_{2.9}$ | $64.7_{1.1}$ | $80.4_{0.2}$ | 2.3 |
| +OU_SDE | $84.7_{0.2}$ | $85.3_{0.1}$ | $\mathbf{92.1}_{0.3}$ | $95.5_{0.2}$ | $\mathbf{80.2}_{0.5}$ | $\mathbf{61.5}_{3.5}$ | $\mathbf{65.9}_{2.7}$ | $\mathbf{80.7}_{0.8}$ | $\mathbf{2.6}$ |
| LoRA | $88.8_{0.1}$ | $89.2_{0.2}$ | $93.5_{0.2}$ | $95.5_{0.1}$ | $84.6_{0.4}$ | $62.8_{1.6}$ | $78.9_{1.6}$ | $84.8_{0.3}$ | - |
| +BROWN_PDF | $\mathbf{88.9}_{0.1}$ | $\mathbf{89.6}_{0.1}$ | $\mathbf{93.9}_{0.1}$ | $95.6_{0.2}$ | $85.1_{0.7}$ | $63.7_{0.5}$ | $80.0_{0.5}$ | $85.2_{0.1}$ | 0.4 |
| +OU_PDF | $\mathbf{88.9}_{0.2}$ | $89.4_{0.1}$ | $93.7_{0.1}$ | $\mathbf{95.7}_{0.3}$ | $86.0_{0.5}$ | $63.6_{0.6}$ | $80.5_{0.8}$ | $85.4_{0.1}$ | 0.6 |
| +BROWN_SDE | $\mathbf{88.9}_{0.1}$ | $89.5_{0.1}$ | $93.7_{0.1}$ | $\mathbf{95.7}_{0.1}$ | $\mathbf{86.5}_{1.2}$ | $\mathbf{63.9}_{0.4}$ | $\mathbf{80.9}_{0.8}$ | $\mathbf{85.6}_{0.1}$ | $\mathbf{0.8}$ |
| +OU_SDE | $88.8_{0.0}$ | $89.5_{0.1}$ | $93.7_{0.2}$ | $\mathbf{95.7}_{0.3}$ | $86.3_{1.2}$ | $63.7_{0.6}$ | $80.1_{0.9}$ | $85.4_{0.1}$ | 0.6 |
| BITFIT | $\mathbf{87.9}_{0.2}$ | $87.6_{0.1}$ | $92.7_{0.2}$ | $95.6_{0.1}$ | $79.4_{2.3}$ | $60.2_{0.8}$ | $77.0_{1.5}$ | $82.9_{0.3}$ | - |
| +BROWN_PDF | $\mathbf{87.9}_{0.1}$ | $\mathbf{87.8}_{0.0}$ | $\mathbf{93.0}_{0.2}$ | $95.7_{0.1}$ | $83.1_{0.8}$ | $60.3_{0.6}$ | $78.3_{0.9}$ | $83.7_{0.2}$ | 0.8 |
| +OU_PDF | $\mathbf{87.9}_{0.1}$ | $\mathbf{87.8}_{0.1}$ | $\mathbf{93.0}_{0.1}$ | $95.7_{0.1}$ | $82.6_{0.8}$ | $59.8_{1.0}$ | $\mathbf{78.8}_{1.4}$ | $83.6_{0.4}$ | 0.7 |
| +BROWN_SDE | $\mathbf{87.9}_{0.2}$ | $87.7_{0.0}$ | $92.8_{0.1}$ | $95.7_{0.1}$ | $83.3_{0.8}$ | $\mathbf{61.1}_{1.2}$ | $77.7_{1.5}$ | $\mathbf{83.8}_{0.3}$ | $\mathbf{0.9}$ |
| +OU_SDE | $\mathbf{87.9}_{0.1}$ | $87.6_{0.1}$ | $92.7_{0.1}$ | $\mathbf{95.8}_{0.1}$ | $\mathbf{83.7}_{0.8}$ | $60.8_{1.6}$ | $77.1_{0.9}$ | $83.7_{0.4}$ | 0.8 |
| ADAPTER | $88.8_{0.1}$ | $89.6_{0.3}$ | $93.7_{0.1}$ | $95.6_{0.1}$ | $83.6_{0.1}$ | $60.4_{1.2}$ | $79.5_{1.2}$ | $84.5_{0.3}$ | - |
| +BROWN_PDF | $\mathbf{89.0}_{0.1}$ | $89.7_{0.2}$ | $93.8_{0.2}$ | $\mathbf{95.8}_{0.1}$ | $86.5_{1.1}$ | $\mathbf{62.6}_{0.7}$ | $\mathbf{83.2}_{0.2}$ | $\mathbf{85.8}_{0.2}$ | $\mathbf{1.3}$ |
| +OU_PDF | $88.9_{0.1}$ | $89.7_{0.0}$ | $93.8_{0.1}$ | $\mathbf{95.8}_{0.1}$ | $\mathbf{86.8}_{0.6}$ | $61.9_{0.2}$ | $82.0_{0.6}$ | $85.6_{0.1}$ | 1.1 |
| +BROWN_SDE | $88.9_{0.1}$ | $\mathbf{89.8}_{0.1}$ | $\mathbf{93.9}_{0.2}$ | $\mathbf{95.8}_{0.1}$ | $85.9_{0.4}$ | $62.3_{1.8}$ | $82.2_{0.2}$ | $85.5_{0.2}$ | 1.0 |
| +OU_SDE | $88.9_{0.1}$ | $\mathbf{89.8}_{0.1}$ | $93.7_{0.1}$ | $95.7_{0.1}$ | $85.9_{0.7}$ | $62.5_{1.2}$ | $82.7_{0.5}$ | $85.6_{0.2}$ | 1.1 |

Table 8: The results on GLUE for Deberta$_{\text{xlarge}}$.

| PET | MNLI | QQP | QNLI | SST-2 | MRPC | CoLA | RTE | Average | Δ |
|---|---|---|---|---|---|---|---|---|---|
| PROMPT | $87.2_{0.1}$ | $86.5_{0.1}$ | $93.8_{0.1}$ | $\mathbf{96.8}_{0.1}$ | $75.4_{2.5}$ | $64.2_{3.8}$ | $78.8_{3.5}$ | $83.2_{0.5}$ | - |
| +BROWN_PDF | $\mathbf{87.6}_{0.1}$ | $\mathbf{86.8}_{0.3}$ | $\mathbf{94.2}_{0.1}$ | $\mathbf{96.8}_{0.1}$ | $80.3_{2.9}$ | $\mathbf{65.5}_{1.3}$ | $\mathbf{79.8}_{0.5}$ | $84.5_{0.6}$ | 1.3 |
| +BROWN_SDE | $\mathbf{87.6}_{0.1}$ | $\mathbf{86.8}_{0.1}$ | $94.0_{0.1}$ | $\mathbf{96.8}_{0.1}$ | $\mathbf{84.4}_{0.6}$ | $64.8_{0.5}$ | $79.5_{1.3}$ | $\mathbf{84.8}_{0.2}$ | $\mathbf{1.6}$ |
| LoRA | $\mathbf{91.1}_{0.1}$ | $90.3_{0.1}$ | $95.1_{0.1}$ | $96.8_{0.1}$ | $88.7_{0.7}$ | $68.0_{1.3}$ | $83.4_{1.1}$ | $87.6_{0.3}$ | - |
| +BROWN_PDF | $\mathbf{91.1}_{0.0}$ | $\mathbf{90.5}_{0.0}$ | $\mathbf{95.2}_{0.0}$ | $\mathbf{97.0}_{0.1}$ | $90.1_{0.8}$ | $68.6_{0.8}$ | $\mathbf{85.9}_{1.3}$ | $88.3_{0.2}$ | 0.7 |
| +BROWN_SDE | $\mathbf{91.1}_{0.1}$ | $90.4_{0.0}$ | $95.1_{0.0}$ | $96.9_{0.2}$ | $\mathbf{90.5}_{0.6}$ | $\mathbf{69.6}_{1.1}$ | $85.6_{0.9}$ | $\mathbf{88.5}_{0.1}$ | 0.9 |
| BITFIT | $90.0_{0.1}$ | $\mathbf{88.4}_{0.0}$ | $\mathbf{95.0}_{0.0}$ | $\mathbf{96.6}_{0.1}$ | $87.3_{0.6}$ | $66.9_{0.2}$ | $82.4_{0.6}$ | $86.7_{0.1}$ | - |
| +BROWN_PDF | $\mathbf{90.2}_{0.0}$ | $88.3_{0.1}$ | $\mathbf{95.0}_{0.1}$ | $\mathbf{96.6}_{0.1}$ | $89.8_{0.5}$ | $\mathbf{67.9}_{0.8}$ | $82.9_{0.6}$ | $87.2_{0.1}$ | 0.5 |
| +BROWN_SDE | $90.1_{0.1}$ | $88.3_{0.0}$ | $94.8_{0.0}$ | $\mathbf{96.6}_{0.1}$ | $\mathbf{90.4}_{0.5}$ | $\mathbf{67.9}_{0.4}$ | $\mathbf{83.8}_{0.5}$ | $\mathbf{87.4}_{0.1}$ | $\mathbf{0.7}$ |
| ADAPTER | $91.1_{0.1}$ | $90.0_{0.1}$ | $95.2_{0.0}$ | $96.8_{0.2}$ | $87.9_{0.5}$ | $68.8_{1.8}$ | $85.0_{0.6}$ | $87.8_{0.2}$ | - |
| +BROWN_PDF | $\mathbf{91.2}_{0.1}$ | $90.0_{0.0}$ | $\mathbf{95.3}_{0.1}$ | $\mathbf{96.9}_{0.2}$ | $89.2_{0.8}$ | $70.1_{1.0}$ | $\mathbf{86.9}_{1.5}$ | $88.5_{0.5}$ | 0.7 |
| +BROWN_SDE | $\mathbf{91.2}_{0.2}$ | $\mathbf{90.1}_{0.1}$ | $95.2_{0.0}$ | $\mathbf{96.9}_{0.2}$ | $\mathbf{90.3}_{0.9}$ | $\mathbf{70.8}_{1.1}$ | $86.3_{1.4}$ | $\mathbf{88.7}_{0.4}$ | $\mathbf{0.9}$ |

Table 9: The results on GLUE for Deberta$_{\text{xlarge}}$ under 16-shot setting.

| PET | MNLI | QQP | QNLI | SST-2 | MRPC | CoLA | RTE | Average | Δ |
|---|---|---|---|---|---|---|---|---|---|
| PROMPT | $34.4_{1.2}$ | $53.2_{5.1}$ | $51.7_{1.7}$ | $73.3_{8.7}$ | $50.2_{3.2}$ | $2.5_{2.5}$ | $52.2_{4.9}$ | $45.4_{2.1}$ | - |
| +BROWN_PDF | $35.8_{0.9}$ | $57.7_{2.4}$ | $\mathbf{53.5}_{1.5}$ | $\mathbf{87.5}_{3.0}$ | $52.2_{1.9}$ | $2.8_{2.9}$ | $\mathbf{54.1}_{1.8}$ | $\mathbf{49.1}_{0.8}$ | $\mathbf{3.7}$ |
| +BROWN_SDE | $\mathbf{35.9}_{1.6}$ | $\mathbf{59.6}_{6.4}$ | $53.1_{1.9}$ | $82.1_{4.2}$ | $\mathbf{55.4}_{1.4}$ | $\mathbf{2.9}_{3.0}$ | $\mathbf{54.1}_{1.3}$ | $49.0_{1.1}$ | 3.6 |
| LoRA | $43.1_{3.6}$ | $68.4_{2.9}$ | $60.1_{5.3}$ | $\mathbf{91.8}_{1.1}$ | $57.6_{1.5}$ | $3.1_{3.8}$ | $56.6_{2.3}$ | $54.4_{1.0}$ | - |
| +BROWN_PDF | $\mathbf{52.1}_{3.4}$ | $70.2_{2.9}$ | $\mathbf{73.3}_{6.3}$ | $91.7_{1.1}$ | $59.5_{4.5}$ | $14.2_{5.6}$ | $60.4_{4.2}$ | $60.2_{1.2}$ | 5.8 |
| +BROWN_SDE | $49.6_{4.3}$ | $\mathbf{70.6}_{1.5}$ | $72.4_{6.0}$ | $90.7_{1.2}$ | $\mathbf{59.8}_{4.5}$ | $\mathbf{28.9}_{1.5}$ | $\mathbf{60.6}_{3.1}$ | $\mathbf{61.8}_{0.5}$ | $\mathbf{7.4}$ |
| BITFIT | $41.9_{3.8}$ | $67.7_{2.6}$ | $60.3_{4.2}$ | $\mathbf{91.8}_{0.8}$ | $54.9_{2.5}$ | $9.4_{2.4}$ | $57.6_{2.0}$ | $54.8_{0.8}$ | - |
| +BROWN_PDF | $45.2_{3.7}$ | $\mathbf{70.3}_{1.2}$ | $65.4_{6.7}$ | $90.9_{0.8}$ | $55.6_{2.1}$ | $8.2_{2.5}$ | $\mathbf{59.6}_{2.3}$ | $56.5_{0.7}$ | 1.7 |
| +BROWN_SDE | $\mathbf{45.7}_{3.8}$ | $69.2_{2.3}$ | $\mathbf{69.2}_{6.3}$ | $89.7_{1.3}$ | $\mathbf{57.8}_{4.0}$ | $\mathbf{24.2}_{4.6}$ | $59.4_{2.8}$ | $\mathbf{59.3}_{1.2}$ | $\mathbf{4.5}$ |
| ADAPTER | $43.1_{2.9}$ | $67.7_{2.7}$ | $55.9_{5.3}$ | $\mathbf{91.1}_{0.9}$ | $56.1_{2.1}$ | $8.6_{5.6}$ | $59.0_{2.3}$ | $54.5_{1.4}$ | - |
| +BROWN_PDF | $\mathbf{50.7}_{3.0}$ | $70.1_{1.6}$ | $70.9_{5.2}$ | $90.6_{1.7}$ | $57.6_{4.3}$ | $16.1_{7.4}$ | $\mathbf{60.6}_{3.9}$ | $59.5_{0.8}$ | 5.0 |
| +BROWN_SDE | $47.1_{1.6}$ | $\mathbf{72.0}_{1.1}$ | $\mathbf{71.3}_{4.3}$ | $91.0_{1.0}$ | $\mathbf{59.7}_{4.5}$ | $\mathbf{26.4}_{5.5}$ | $60.0_{5.7}$ | $\mathbf{61.1}_{0.6}$ | $\mathbf{6.6}$ |

Table 10: Hyper-parameters for training $g_\gamma$

| | PDF | SDE |
|---|---|---|
| Learning rate | 1e-3 | 1e-3 |
| Weight decay | 0 | 0 |
| Batch size | 128 | 128 |
| Grad norm | 1.0 | 1.0 |
| Max steps | 100k | 500k |
| Warmup ratio | 0.01 | 0.01 |

### E.2   TRAINING PETs ON FULL-SET GLUE

We run all the experiments for 50k steps, and evaluate on the development set every 1k steps. For BERT$_{\text{large}}$, we use 32 as the batch size while for Deberta$_{\text{xlarge}}$, we use 16 as the batch size. We choose learning rate 1e-3 for Prompt tuning for both PLMs, and 1e-4 for other PETs for both PLMs. We use 0.01 weight decay, 1.0 maximum gradient norm and no learning rate warm-up for all the experiments. We search the best regularization strength $\alpha$ in {0.01, 0.05, 0.1, 0.2, 0.3, 0.4, 0.5, 0.6, 0.7, 0.8, 0.9, 1.0} for PDF regularizer, and in {0.0001, 0.0005, 0.001, 0.005, 0.01, 0.05, 0.1, 0.2, 0.5, 1.0} for SDE regularizer. The best $\alpha$ for PDF regularizer are listed in Table 11, and best $\alpha$ for SDE regularizer are listed in Table 12.

Table 11: Best $\alpha$ for PDF regularizer on full-set GLUE

|  | MNLI | QQP | QNLI | SST-2 | MRPC | CoLA | RTE |
|---|---|---|---|---|---|---|---|
| *BERT$_{large}$* | | | | | | | |
| PROMPT | 0.05 | 0.3 | 0.01 | 0.2 | 0.8 | 0.4 | 0.4 |
| LORA | 0.2 | 0.8 | 0.5 | 0.1 | 0.1 | 0.5 | 0.3 |
| BIAS | 0.05 | 0.3 | 0.6 | 0.05 | 0.8 | 0.05 | 0.7 |
| ADAPTER | 0.8 | 0.01 | 0.7 | 0.1 | 0.6 | 0.4 | 0.3 |
| *Deberta$_{xlarge}$* | | | | | | | |
| PROMPT | 0.05 | 0.1 | 0.2 | 0.8 | 0.7 | 0.3 | 0.7 |
| LORA | 0.1 | 0.1 | 0.05 | 0.01 | 0.1 | 0.01 | 0.2 |
| BIAS | 0.05 | 0.01 | 0.01 | 0.01 | 0.8 | 0.01 | 0.8 |
| ADAPTER | 0.3 | 0.01 | 0.01 | 0.01 | 0.01 | 0.6 | 0.1 |

Table 12: Best $\alpha$ for SDE regularizer on full-set GLUE

|  | MNLI | QQP | QNLI | SST-2 | MRPC | CoLA | RTE |
|---|---|---|---|---|---|---|---|
| *BERT$_{large}$* | | | | | | | |
| PROMPT | 0.005 | 0.005 | 0.0005 | 0.0001 | 1.0 | 0.2 | 0.05 |
| LORA | 0.0005 | 0.01 | 0.01 | 0.01 | 0.005 | 0.0005 | 0.0005 |
| BIAS | 0.001 | 0.001 | 0.005 | 0.0001 | 1.0 | 1.0 | 0.005 |
| ADAPTER | 0.001 | 0.001 | 0.0005 | 0.001 | 0.005 | 0.5 | 0.005 |
| *Deberta$_{xlarge}$* | | | | | | | |
| PROMPT | 0.0001 | 0.001 | 0.0001 | 0.0001 | 0.2 | 0.0001 | 0.0001 |
| LORA | 0.0005 | 0.05 | 0.0001 | 0.0001 | 0.1 | 0.005 | 0.0001 |
| BIAS | 0.0001 | 0.0005 | 0.0001 | 0.0001 | 0.05 | 0.2 | 0.001 |
| ADAPTER | 0.001 | 0.0005 | 0.0001 | 0.0005 | 0.1 | 0.001 | 0.0005 |

### E.3 TRAINING PETs ON FEW-SHOT GLUE

We run all the experiments for 1k steps, and evaluate on the development set every 50 steps. For all the shots for both regularizers and both models, we use a batch size of 2. Other hyper-parameters are kept the same as in the experiments on full-set GLUE.

## F PERFORMANCE OF REGULARIZERS TRAINED WITH TINY CORPUS

Although we use the pre-training corpus to train our mapping $g_\gamma$, the training is actually fast and data-efficient. We show that when using only 10,000 documents in the pre-training corpus (about 0.1% of the corpus), the obtained regularizers still perform great and are comparable to the regularizers trained on the whole pre-training corpus. We train the mapping $g_\gamma$ for 5,000 iterations with 128 batch size. On a single NVIDIA A100 GPU, the training can be done in 1 hour for PDF regularizer, and 3 hours for SDE regularizer. The cost of training our regularizer is quite small compared to the resources required for pre-training.

We conduct the same experiments as Section 4.2 and Section 4.3 with the regularizers trained with tiny corpus. The results are presented in Table 13 and Table 14 respectively.

On full-set GLUE, the PDF regularizer performs even better on three out of four PETs, and although its performance is affected on Adapter, it still outperforms the vanilla Adapter. The SDE regularizer is slightly affected on three out of four PETs, but it still brings substantial improvements on all the PETs.

On few-shot GLUE, the impact of the shrinkage of the corpus is relatively obvious. But overall, the regularizers still performs great on all the PETs. The drop in performances are relatively small compared to the boost they bring to vanilla PETs.

Table 13: The results on GLUE for BERT$_{large}$ *with regularizers trained on 0.1% of the pre-training corpus.* $\Delta_{whole}$ is the difference between the the average performance in this table and the average performance in Table 1.

| PET | MNLI | QQP | QNLI | SST-2 | MRPC | CoLA | RTE | Average | $\Delta$ | $\Delta_{whole}$ |
|---|---|---|---|---|---|---|---|---|---|---|
| PROMPT | $84.4_{0.1}$ | $85.3_{0.3}$ | $91.5_{0.1}$ | $95.5_{0.1}$ | $73.9_{2.4}$ | $55.5_{3.4}$ | $60.8_{1.5}$ | $78.1_{0.6}$ | - | - |
| +BROWN_PDF | $84.7_{0.0}$ | $\mathbf{85.4}_{0.1}$ | $\mathbf{92.1}_{0.1}$ | $\mathbf{95.8}_{0.2}$ | $75.8_{1.4}$ | $56.6_{0.9}$ | $61.1_{3.0}$ | $78.8_{0.5}$ | 0.7 | +0.1 |
| +BROWN_SDE | $\mathbf{84.8}_{0.2}$ | $\mathbf{85.4}_{0.1}$ | $\mathbf{92.1}_{0.1}$ | $\mathbf{95.8}_{0.2}$ | $\mathbf{79.0}_{1.3}$ | $\mathbf{59.8}_{8.2}$ | $\mathbf{65.5}_{1.2}$ | $\mathbf{80.3}_{1.2}$ | 2.2 | -0.1 |
| LoRA | $88.8_{0.1}$ | $89.2_{0.2}$ | $93.5_{0.2}$ | $95.5_{0.1}$ | $84.6_{0.4}$ | $62.8_{1.6}$ | $78.9_{1.6}$ | $84.8_{0.3}$ | - | - |
| +BROWN_PDF | $\mathbf{88.9}_{0.1}$ | $\mathbf{89.5}_{0.1}$ | $\mathbf{93.8}_{0.1}$ | $\mathbf{95.8}_{0.2}$ | $86.0_{0.6}$ | $\mathbf{64.4}_{0.6}$ | $80.4_{0.3}$ | $\mathbf{85.5}_{0.1}$ | 0.7 | +0.3 |
| +BROWN_SDE | $\mathbf{88.9}_{0.0}$ | $89.4_{0.1}$ | $\mathbf{93.8}_{0.1}$ | $95.6_{0.2}$ | $\mathbf{86.8}_{0.5}$ | $63.5_{0.9}$ | $80.9_{0.8}$ | $\mathbf{85.5}_{0.2}$ | 0.7 | -0.1 |
| BITFIT | $\mathbf{87.9}_{0.2}$ | $87.6_{0.1}$ | $92.7_{0.2}$ | $95.6_{0.1}$ | $79.4_{2.3}$ | $60.2_{0.8}$ | $77.0_{1.5}$ | $82.9_{0.3}$ | - | - |
| +BROWN_PDF | $\mathbf{87.9}_{0.1}$ | $\mathbf{87.7}_{0.1}$ | $\mathbf{92.8}_{0.1}$ | $95.7_{0.2}$ | $\mathbf{83.6}_{0.7}$ | $\mathbf{61.2}_{0.2}$ | $\mathbf{78.7}_{1.1}$ | $\mathbf{84.0}_{0.1}$ | 1.1 | +0.3 |
| +BROWN_SDE | $\mathbf{87.9}_{0.1}$ | $\mathbf{87.7}_{0.1}$ | $\mathbf{92.8}_{0.1}$ | $95.7_{0.1}$ | $82.8_{0.7}$ | $61.1_{0.8}$ | $77.4_{0.6}$ | $83.6_{0.3}$ | 0.7 | -0.2 |
| ADAPTER | $88.8_{0.1}$ | $89.6_{0.3}$ | $93.7_{0.1}$ | $95.6_{0.1}$ | $83.6_{0.1}$ | $60.4_{1.2}$ | $79.5_{1.2}$ | $84.5_{0.3}$ | - | - |
| +BROWN_PDF | $\mathbf{88.9}_{0.1}$ | $89.8_{0.1}$ | $\mathbf{93.8}_{0.2}$ | $\mathbf{95.9}_{0.2}$ | $85.4_{0.8}$ | $60.9_{0.5}$ | $\mathbf{82.7}_{1.8}$ | $85.3_{0.2}$ | 0.8 | -0.5 |
| +BROWN_SDE | $\mathbf{88.9}_{0.0}$ | $89.8_{0.1}$ | $93.7_{0.1}$ | $95.8_{0.3}$ | $\mathbf{85.7}_{0.5}$ | $\mathbf{61.9}_{0.9}$ | $82.4_{0.3}$ | $\mathbf{85.5}_{0.2}$ | 1.0 | 0.0 |

Table 14: The results on GLUE for BERT$_{large}$ under 16-shot setting *with regularizers trained on 0.1% of the pre-training corpus.* $\Delta_{whole}$ is the difference between the the average performance in this table and the average performance in Table 2.

| PET | MNLI | QQP | QNLI | SST-2 | MRPC | RTE | Average | $\Delta$ | $\Delta_{whole}$ |
|---|---|---|---|---|---|---|---|---|---|
| PROMPT | $38.1_{1.5}$ | $53.0_{3.1}$ | $51.6_{1.4}$ | $70.1_{4.9}$ | $50.1_{3.0}$ | $48.0_{1.3}$ | $51.8_{0.9}$ | - | - |
| +BROWN_PDF | $38.4_{1.5}$ | $54.6_{2.5}$ | $52.4_{1.4}$ | $73.1_{6.0}$ | $\mathbf{54.0}_{1.4}$ | $\mathbf{51.1}_{1.7}$ | $53.9_{1.1}$ | 2.1 | -0.1 |
| +BROWN_SDE | $\mathbf{39.3}_{1.1}$ | $\mathbf{56.2}_{3.0}$ | $\mathbf{52.8}_{1.1}$ | $\mathbf{80.8}_{7.1}$ | $53.0_{6.0}$ | $51.0_{2.5}$ | $\mathbf{55.5}_{0.9}$ | 3.7 | +0.1 |
| LoRA | $48.7_{4.5}$ | $59.9_{5.5}$ | $53.2_{1.2}$ | $90.2_{1.1}$ | $53.6_{3.4}$ | $64.2_{0.9}$ | $61.6_{0.6}$ | - | - |
| +BROWN_PDF | $51.4_{2.2}$ | $62.0_{1.8}$ | $54.8_{3.1}$ | $\mathbf{91.2}_{0.3}$ | $57.4_{4.0}$ | $65.7_{0.7}$ | $63.7_{0.9}$ | 2.1 | -0.3 |
| +BROWN_SDE | $\mathbf{53.2}_{1.9}$ | $\mathbf{65.4}_{1.7}$ | $\mathbf{64.2}_{5.0}$ | $\mathbf{91.2}_{0.3}$ | $\mathbf{61.4}_{3.6}$ | $\mathbf{66.3}_{0.7}$ | $\mathbf{66.9}_{0.7}$ | 5.3 | -0.1 |
| BITFIT | $48.4_{1.6}$ | $56.0_{6.1}$ | $51.7_{2.5}$ | $90.8_{0.8}$ | $52.0_{2.3}$ | $61.7_{1.4}$ | $60.1_{1.1}$ | - | - |
| +BROWN_PDF | $49.1_{1.8}$ | $56.0_{6.0}$ | $53.1_{2.4}$ | $\mathbf{91.1}_{0.2}$ | $54.0_{2.0}$ | $62.8_{1.0}$ | $61.0_{0.7}$ | 0.9 | 0.0 |
| +BROWN_SDE | $\mathbf{51.4}_{1.4}$ | $\mathbf{60.1}_{1.7}$ | $\mathbf{56.7}_{1.8}$ | $\mathbf{91.1}_{0.3}$ | $\mathbf{57.3}_{4.5}$ | $\mathbf{63.0}_{1.2}$ | $\mathbf{63.2}_{0.7}$ | 3.1 | -0.4 |
| ADAPTER | $47.4_{3.7}$ | $57.0_{7.2}$ | $55.8_{2.9}$ | $91.0_{0.4}$ | $55.8_{2.5}$ | $62.7_{2.0}$ | $61.6_{1.2}$ | - | - |
| +BROWN_PDF | $48.1_{4.0}$ | $58.3_{6.9}$ | $57.9_{3.5}$ | $91.4_{0.4}$ | $57.7_{3.6}$ | $63.0_{3.0}$ | $62.7_{1.0}$ | 1.1 | 0.0 |
| +BROWN_SDE | $\mathbf{50.3}_{2.2}$ | $\mathbf{61.0}_{6.1}$ | $\mathbf{63.4}_{4.4}$ | $\mathbf{91.5}_{0.3}$ | $\mathbf{58.4}_{2.2}$ | $\mathbf{64.3}_{1.9}$ | $\mathbf{64.8}_{1.1}$ | 3.2 | -0.6 |

## G SPEED OF THE REGULARIZERS

As we have mentioned in Section 1, the PDF regularizer only incurs negligible computational cost. In this section, we present the plot of step-metric curve on full-set GLUE in Figures 5, 6, 7 and 8.

On different PETs, the regularized PETs with PDF regularizer has similar running time to the vanilla PETs. On the two large datasets, QQP and MNLI, regularized PETs with SDE regularizer take about 2 to 3 times longer to achieve the best performance than vanilla PETs. However, on medium-sized (QNLI, SST-2) and small datasets (CoLA, MRPC, RTE), the time to achieve the best results with SDE regularizer is comparable to vanilla PETs.

Overall, the PDF regularizer can effectively improve the performance of PETs without introducing much computational cost. In scenarios where there is relatively more focus on the inference performance of PETs and less concern about the slightly longer training time, or when the dataset is small, SDE regularizer should be a good choice.

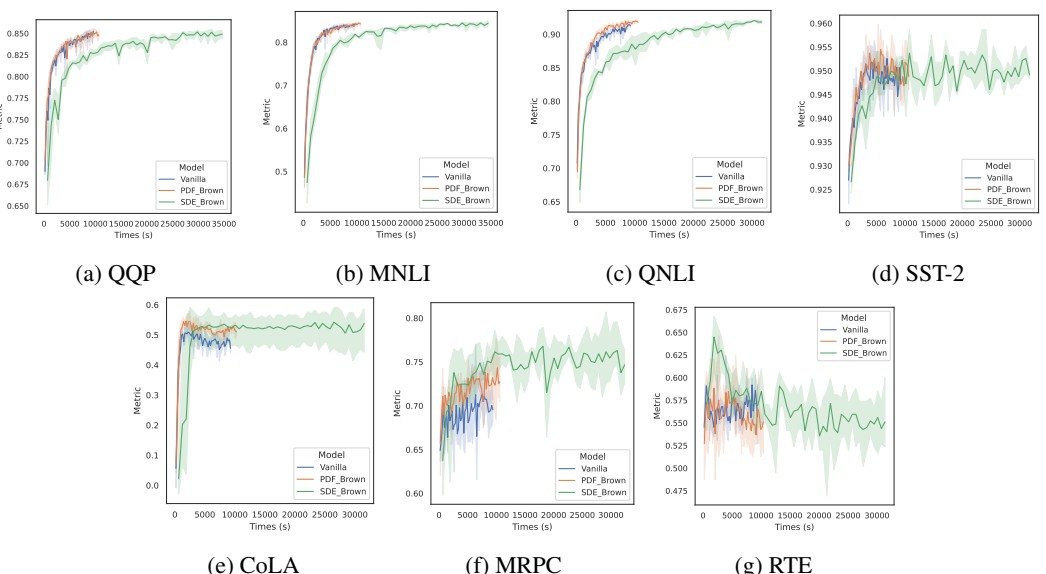

Figure 5: Time-Metric curve for regularizers on *prompt tuning*.

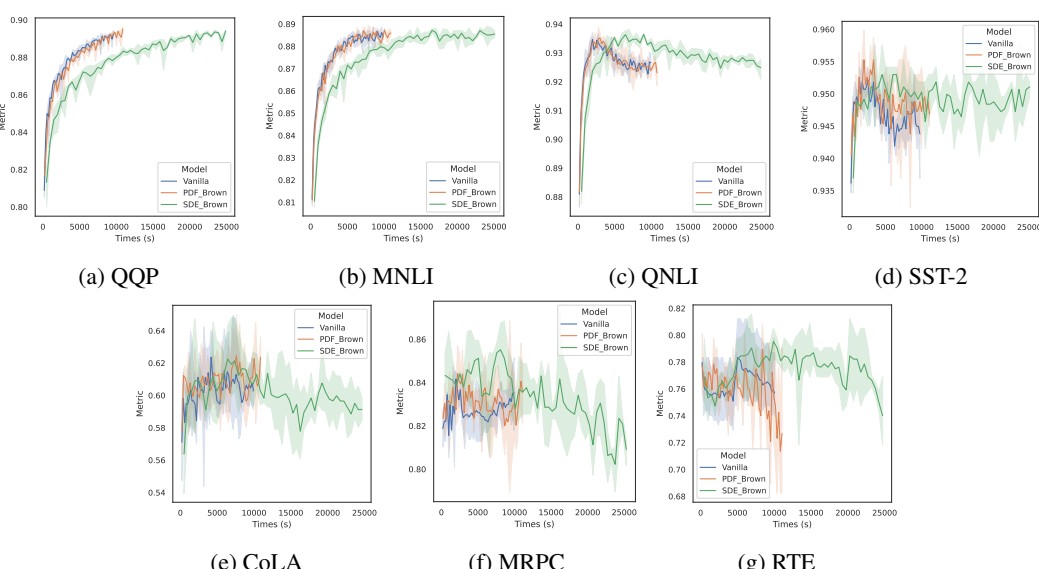

Figure 6: Time-Metric curve for regularizers on *LoRA*.

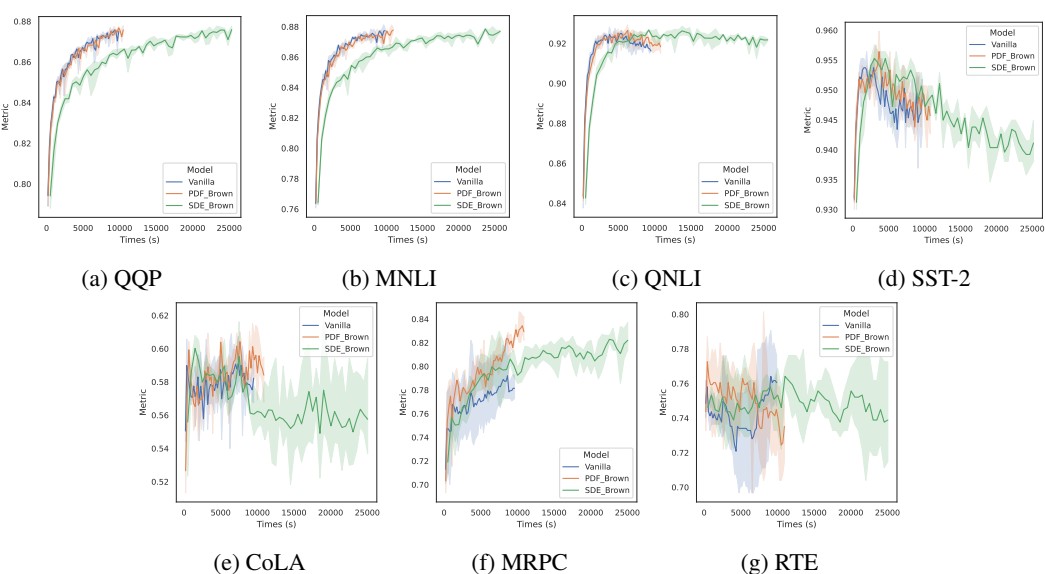

(a) QQP  (b) MNLI  (c) QNLI  (d) SST-2

(e) CoLA  (f) MRPC  (g) RTE

Figure 7: Time-Metric curve for regularizers on *BitFit*.

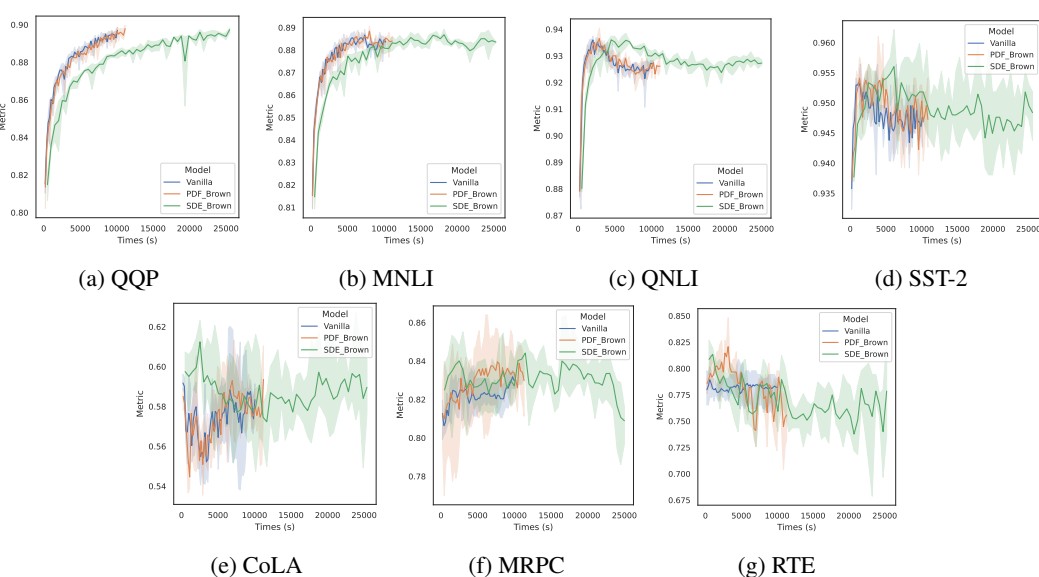

(a) QQP  (b) MNLI  (c) QNLI  (d) SST-2

(e) CoLA  (f) MRPC  (g) RTE

Figure 8: Time-Metric curve for regularizers on *Adapter*.

