# OpenReview forum: "Stochastic Bridges as Effective Regularizers for Parameter-Efficient Tuning"
_ICLR.cc/2023/Conference — Submitted to ICLR 2023_

### Official Review · Reviewer_69ms · 2022-10-23

**Confidence:** 3
**Correctness:** 4
**Technical Novelty And Significance:** 3
**Empirical Novelty And Significance:** 2
**Recommendation:** 6

**Clarity, Quality, Novelty And Reproducibility:**

The paper is well-written in general. The proposed regularizer is a clever integration of language models with diffusion bridges of the target output. Training hyperparameters are included in the appendix.

**Strength And Weaknesses:**

Strength:

- The diffusion bridge forces the latent representation of each layer to contain more information of the target, and can be viewed as a shortcut for learning better latent representations tailored to different tasks.

- The authors demonstrate that task performance is correlated with the distance to the diffusion bridge, which serves as an empirical evidence of the effectiveness of the proposed regularizer.

Weaknesses and questions:

- One weakness of the proposed method is that pre-training corpus is required to train the mapping $g_{\gamma}$. The problem here is two-fold: (i) pre-training corpus could be hard to access and (ii) since it is much larger than datasets of downstream tasks, training $g_{\gamma}$ could be computational demanding. For example, I would like to know whether the proposed method is still effective if we only have access to a small subset of the pre-training corpus.

- Following the previous comment, the computation cost of training $g_{\gamma}$ should be discussed. Furthermore, it would be nice if the authors can provide additional experiments by constraining the GPU time of the baselines and the proposed approach to be the same.

**Summary Of The Paper:**

This paper proposes a class of regularization objectives that are used to improve the effectiveness of parameter-efficient tuning methods (PETs) for pre-trained language models. Specifically, the objective consists of two main components: (i) a (potentially learnable) diffusion bridge defining a target diffusion process; (ii) a mapping function from every layer's output of the language model to a trajectory that approximates the diffusion bridge. This paper uses a fixed diffusion bridge defined by applying PCA to the output word embeddings. Experiment results show that the proposed regularizer is able to improve the performance of existing PETs.

**Summary Of The Review:**

In summery, I tend to vote for acceptance since the proposed regularizer makes clever use of target diffusion processes to improve the fine-tuning performance. However, the proposed method requires pre-training corpus, which could add significant computation cost. It is also not well discussed how much pre-training corpus is needed to receive significant performance gain.

---

> ### Author Response · Authors · 2022-11-18
> **Response to Reviewer 69ms**
>
> We sincerely appreciate your reviews of our work! Regarding the several questions you raised, we reply here one by one.
>
> *Q1: Pre-training corpus could be hard to access & How about the effectiveness of our method when we only have access to a small subset of the pre-training corpus*
>
> A1: It is mostly answered in Q2 of our response to reviewer *PqBu.* In addition, **training the mapping does not take much time**. The training can be done in 5,000 steps with a batch size of 128. For PDF regularizer, **it only takes less than 1 hour to train on a single A100**, which is even faster than training a PET on MNLI (around 2 hours). For SDE regularizer, it is about 3 times slower, but the training time is still acceptable. We add an experiment using only around **0.1%** of the pre-training corpus (10,000 documents) to pre-train the regularizer, the performances of the regularized PETs are either barely affected or even better. See Appendix F of our rebuttal revision.
>
>
>
> *Q2: Experimental results when* *GPU* *time is constrained.*
>
> A2: **Thanks for your advice! We plot the time-metric curve in Appendix G of our rebuttal revision.** Generally, regularized PETs with the PDF regularizer have nearly the same running time as vanilla PETs. PETs with the SDE regularizer converge generally 2-3x slower than vanilla PETs on large datasets such as MNLI and QQP, but on other medium-sized and small datasets, the convergence speeds are comparable to vanilla PETs.
>
>
>
> We ask you to kindly check our rebuttal revision, where we improve our writing and add several experiments according to your and other reviewers’ suggestions. Once again, thank you for your constructive suggestions!

---

### Official Review · Reviewer_CmgL · 2022-11-01

**Confidence:** 2
**Correctness:** 3
**Technical Novelty And Significance:** 2
**Empirical Novelty And Significance:** 2
**Recommendation:** 5

**Clarity, Quality, Novelty And Reproducibility:**

The idea of interpreting interpreting the tunable parameters are control variables on top of a frozen PLM is not novel to this work, as the authors cite (Yang & Liu' 22, Ding et al. '22).

Clarity is good in some aspects, but needs improvements in several other critical components -- for example, the formalism was incomplete as to where the continuous time aspect of the diffusion bridges is entering the regularization framework in the fine-tuning objective.

When $\bar{h}_o^{(i)}$ is defined as "the average of all hidden states at the $i-$th layer",  what does this mean? Doesn't that make it independent of the position $o$ if interpreting "all" as a reduction over the time steps?

What is the $t_i$ in Equations (5) and (6)? Is this just a uniformly spaced grid with length equal to the number of layers? It would be good to make this clear without expecting the reader to guess things.

The diffusion bridge formalism starts at 0 and then ends at a parameter $\beta$. What does this map to for the bridge $X_y$ used in the application (just above Equation (5)? Inspite of having a paragraph Section 3.2 on this, this aspect is still very unclear.

The Method 2 (approximating the SDE)'s description leaves several variables undefined which is concerning given that it is the best performing method in the experiment. Things like $\sigma$, $f^{T; \beta}$ ? There is also an $\tilde{X}_t$, which is supposed to be an estimate that's again undefined.


**Strength And Weaknesses:**

Strengths:

The key idea is to learn a frozen non-linear dimensionality reduction map that recovers the dynamics of a stochastic bridge on the pre-training corpus and then using it in the fine-tuning stage to define an additional regularization cost. Overall, the proposal of using a diffusion bridge with an endpoint derived from the word embedding at the final layer in PLM (a projected version of it) seems novel / interesting.  Justifying the novel fine-tuning objective as a regularization cost seems intuitively plausible, even if the other motivations like considering it as running cost versus terminal cost appear to be a superficial connection. The authors also conduct experiments to show that this scheme can help marginally improve the final results in practice. In addition to evaluating the benchmarks, the authors also conduct some analysis of the hidden states of the various methods in terms of the clustering properties of a reduced dimension visualization.

Weaknesses:

- Clarity of the writing is quite unclear and leaves a lot of details to imagination.  Please see the next section for concrete suggestions.
-  The authors claim that they "reemphasize" a "running cost" in addition to the terminal cost (which is simply the model loss), however it is unclear why this running cost is important for the overall objective, especially given that this is a cost that the authors define, rather than inspired or proposed from a pre-existing consideration or loss.
- It also seems somewhat arbitrary that this new regularization cost is specific to parameter efficient tuning, but it does not seem specific to this setting (e.g. what aspect of going from general finetuning to PET is specific to the method proposed here?)
- Optimal control formulation is often mentioned for the motivation, but the connection is limited to that and not really used in the technical contents of the paper.
- The significance of hidden state in vanilla PETS "approaching the latent bridges" is unclear to me. Specifically, given that the final reduced dimension representations of the embeddings are estimated from the pre-trained corpus, wouldn't a better fit to the end points automatically imply a better prediction accuracy?



**Summary Of The Paper:**



This paper proposes a novel framework to do parameter efficient fine-tuning method for PLM (pre-trained language models). The basic idea is to hypothesize that there exists a map from the hidden states in each layer to a lower dimensional latent space, where the trajectory of this lower dimensional map from the zeroth layer to the final layer is from a diffusion bridge that ends at the PCA reduced dimension points of each output target word in the final layer.  The authors propose an interpretation of this as the running cost to augment the usual final terminal cost of the loss function.


**Summary Of The Review:**

Overall, this paper proposes an interesting idea for a regularizer for fine-tuning a pre-trained language model (PLM). The most interesting part about this is to use the frozen model to constrain the dynamics of the hidden states to follow a distribution similar to those in the pre-training corpus (via a dimensionality reduction). The authors also allude to optimal control connections for this idea, which is not particularly significant in my view. However, the clarity of presentation is still somewhat unclear as I write above in more detail.

The experimental results seem promising, but the claims about a novel discovery related to the hidden states forming a latent bridge might not be fully convincing. Overall, the paper has some merits in terms of novelty and motivation, but the technical details and the significance of the empirical results are not totally convincing.

---

> ### Author Response · Authors · 2022-11-18
> **Response to Reviewer CmgL**
>
> We truly appreciate your patience in pointing out the problems in our writing! We have uploaded our rebuttal revision, in which we follow your suggestions to make ourselves clearer and ensure  formulas and variables are well explained. Please do kindly have a look. Here, we will respond to your concerns.
>
> *Q1: The importance of the running cost is unclear.*
>
> A1: As we have said, when previous works try to formalize PETs as performing optimal control, they did not find a nice equivalent in PETs for the running cost in the optimal control. Therefore, designing a running cost not only makes the optimal control perspective of PETs **theoretically sound**, but also makes PETs **empirically better** (our experiments). The running cost (regularizer) we propose can be seen as providing guidance, telling the model which trajectory could be an efficient path to reach the goal. **The information could be important to PETs since their numbers of parameters are few. And the informative guidance such as the expected trajectory can make the training of PETs easier**. Though our regularizers may not be the best running cost for PETs, we argue that improving PETs using better running cost functions will be of interest to many researchers working on PETs, and is a direction worth exploring.
>
>
>
> *Q2: The regularizer seems to be specific to PETs*
>
> A2: Technically, our regularizer would have no problem applying to fine-tuning, but our assumption is that **the system is fixed** so that its dynamics can be learned and applied to downstream tasks. Also, a fixed system is a requirement for the optimal control perspective to step in. Fine-tuning modifies all the parameters of the PLM, dramatically changing the system. It makes little sense to use the previous dynamics of the system to regularize the new dynamics. Exploring the relation between the change of the system and the change of the dynamics should be an interesting and complex direction to explore. However, it is out of the scope of our paper, and we leave it as our future work.
>
>
>
> *Q3: Optimal control is not really used in our technical contents*
>
> A3: We ask you to kindly refer to Q1 in our response to reviewer PqBu. Also, using more advanced optimal control techniques could be more principled and we will consider it in our future works.
>
>
>
> *Q4: The significance of the hidden states in vanilla PETs "approaching the latent bridges" is unclear*
>
> A4: By saying "approaching the latent bridges", we do not just mean that the endpoints are better fitted, but that the **intermediate states** are also approaching our latent bridge. Although the phenomenon seems intuitive, it is definitely non-trivial. The domains and training objectives of the downstream tasks are different from pre-training, it is very possible that the trained PETs result in completely different dynamics from that on the pre-training corpus (except the end points). What we show is that although there may be potentially infinite paths from the head endpoint to the tail endpoint, the path formed by the intermediate hidden states approaches our latent bridges instead of other paths. This justifies our use of PLM's dynamics as the regularizer for PETs.
>
>
>
> Q5: Clarity-related issues
>
> A5: We answer the clarity issues here. You can refer to our general response for changes we made in the rebuttal revision
>
> 1. *where the continuous time aspect of the diffusion bridges is entering the regularization framework*: For PDF regularizer, time does not necessarily need to be continuous, since it is based on the transition PDF between two time points. For SDE regularizer, time has to be continuous since solving SDE requires continuous representation of the hidden states. We have explicitly distinguished the two cases in Eq.5 and Eq.6 in our revision.
> 2. *$\bar{\mathbf{h}}_o^{(i)}$ is not clear*: We mistakenly added the subscript $o$ to this vector, and we have corrected.
> 3. *What is the $t_i$ in Eq.5 and Eq.6*: Resolved together with the first issue.
> 4. *What is the bridge in application*: Each word has a diffusion bridge that starts at 0 and ends at its low-dimensional word embedding (obtained by PCA). $X_y$ refers to the diffusion bridge starting at 0 and ending at PCA(Embed[y]).
> 5. Several variables in Method 2 are undefined: We have rewritten the formula.

---

### Official Review · Reviewer_PqBu · 2022-11-03

**Confidence:** 4
**Clarity, Quality, Novelty And Reproducibility:** 1. The paper was a little bit hard to…
**Correctness:** 3
**Technical Novelty And Significance:** 3
**Empirical Novelty And Significance:** 2
**Recommendation:** 3

**Strength And Weaknesses:**

Strength
- Since the proposed regularization is not dependent of parameter tuning methods, it can be used for many parameter tuning methods.
- The experiments shows improvements on GLEU benchmarks in both full-set and few-shot scenario.

Weakness
- It was not clear what optimal control perspective implies. It's common to use regularization / prior / pre-training in machine learning but the new perspective implies something special (which lead to the proposed method)?
- The method assumes that the pre-training corpus is available, but previous works do not assume so.
- The method looked like pre-training of newly introduced parameters (effective for few-shot setup). Is there any justifications why the stochastic bridges is necessary?
  - What will be the difference between stochastic bridges and pre-training the new parameters on PLM objective?

**Summary Of The Paper:**

Since recent pre-trained language models are large, it is common to fine-tune small amount of parameters instead of tuning all the parameters when we would like to adapt the pre-trained model for downstream tasks (e.g. text classification etc.). Many methods have been proposed such as Prompt tuning, Adapter, BitFit and LoRa (summarized in appendix A). For example, Adapter, one of the simplest approach inserts extra layers (called adapter layers) in transformer blocks and train those layers with supervision from downstream tasks.

The straight forward approach is to directly optimize the downstream task loss, but instead authors proposed a regularization method. The proposed regularization objectives encourage hidden states of each layer to be able to approximate low-rank vector representation (PCA) of target tokens following the diffusion process explained in sec. 2.2 (and parameterized as in sec. 3). The proposed objective is added to the task objectives with some weights.

They evaluated their regularization method on fine-tuning BERT_large model with four parameter tuning approaches (Prompt tuning, Adapter, BitFit and LoRa) and show that the regularization lead to performance improvements on GLEU benchmarks in both full-set and few-shot scenario.

**Summary Of The Review:**

Although the proposed method show improvements, it requires extra data to pre-train the parameters than baselines. And it is hard to make fair comparison with existing methods especially for the few-shot setup. Also, the use of stochastic bridges is not justified conceptually and empirically. For these reasons, I believe this paper is not ready for publication.

---

> ### Author Response · Authors · 2022-11-18
> **Response to Reviewer PqBu (1/2)**
>
> We sincerely thank you for your careful reading and your constructive comments! We would like to respond to the weaknesses you mentioned one by one. We add the contents in this response to our rebuttal revision to improve our paper. Thanks for raising the issues in our paper. It is very helpful for us to improve our paper. Please kindly have a look at our rebuttal revision.
>
> *Q1: What optimal control perspective implies is unclear.*
>
> A1: The special point of PETs is that previous works trained the whole model, while PETs train some specified / newly introduced parameters while fixing the PLM. This is why the perspective of optimal control can step in — PLM resembles the system, and the parameters of PETs resemble the controls. **Just as the concept of *prior* and *posterior* can be naturally introduced and added to the process of machine learning under the perspective of Bayesian learning, we propose the regularizers on hidden states from the perspective of optimal control**. Most existing regularization methods are targeted at the parameters, while we propose to regularize the trajectory of hidden states. Under the perspective of optimal control, what we argue is novel and natural.
>
> *Q2: The pre-training corpus is assumed to be available, but previous works do not assume so, making the comparison unfair.*
>
> A2: We would like to clarify several points.
> 1. **Our aim is to introduce a plug-and-play universal regularizer in the training of PETs rather than proposing a new PET. The contribution of our work (regularizers for PETs) and previous works (PETs) are orthogonal**. We are not comparing our methods with PETs, but showing that our method can be applied to improve PETs. Besides, using the pre-training corpus to train the regularizers should not be seen as introducing unfairness, see Q3.
> 2. In fact, **the pre-training corpus is relatively easy to obtain now. Almost all the PLMs use publicly available corpora for pre-training**, and thanks to the community's growth, these **corpora** have become very easy to download via toolkits such as Huggingface datasets.
> 3. The researchers who pre-train their own model can additionally train such regularizers on top of their PLM using very little time (see Q1 in our response to reviewer 69ms), and **release the regularizers together with their PLM.** Since the regularizers are plug-and-play universal regularizers for PETs, users in different downstream domains can plug the lightweight regularizers into the PLM without any difficulty and obtain better PETs.
> 4. If the situation is really special, for example, the authors of a certain PLM do not make the pre-training corpus publicly available, **their pre-training corpus is not a strict requirement** when a user of the PLM wants to train a regularizer. A well-trained PLM should perform well on other public generic corpora, and users can train the regularizer on these public generic corpora. In this work, to avoid introducing new information and ensure the fairness of the experiment, we directly use the pre-training corpus (Q3).
> 5. **Even if we only have access to a tiny fraction of the pre-trained corpus, it is enough to train the regularizers**. We add an experiment using only **0.1%** of the pre-training corpus (10,000 documents) to train the regularizers, and show that the results of the regularized PETs are barely affected (see Appendix F of our rebuttal revision).
>
> *Q3:* *We use more data*
>
> A3: We actually do not use more data. We only use the pre-training corpus that the PLM has seen during the pre-training. The use of pre-training corpus in our method is to let regularizers learn the dynamics of PLMs. The corpus is also unavailable during the training of PETs. **No new information is introduced, and no information about the downstream task is leaked**. The models and corpus involved in our process are consistent with vanilla PETs, without using any new data or information.
>
> *Q4: Justification of the necessity of stochastic bridges.*
>
> A4: **Since the dynamics of the PLM should be complicated, and things like stochastic bridges and SDEs are handy and battle-tested tools to describe the dynamics of a complex system in engineering, finance, biology, etc. It is natural for us to select such tools to serve our needs.** We use stochastic bridges as an example to show that adding such regularization is beneficial for PETs. We believe that the optimal design of the regularization is a highly open question that can be left to more future works to explore. **Our experimental results on different PETs, different tasks and different PLMs also show that how to regularize PETs is a direction worth paying attention to**

---

> > ### Author Response · Authors · 2022-11-18
> > **Response to Reviewer PqBu (2/2)**
> >
> > *Q5: Difference between* *using* *stochastic bridges and* *using*  *PLM objectives to pre-train some new parameters.*
> >
> > A5: We think that these two things are completely different. Pre-training the new parameters and adding them to the PLM may increase the model's capability, eventually making PETs perform better. **But we argue that we aim to improve the PETs by regularizing the dynamics of their intermediate hidden states.** Adding parameters and adding regularizations are two different directions to improve the model, and the two methods can even be adopted simultaneously to obtain better performance. Since all parameters of the mapping function that projects the hidden states into the latent space are fixed during training PETs, the learnable parameter spaces of vanilla PETs and regularized PETs are the same. Moreover, as we have mentioned in Section 3.1 (the last sentence), our mapping is not involved in the inference of PETs, meaning that we are not improving the performance of PETs by increasing the number of parameter. **In summary, the regularized PETs and vanilla PETs have the same architecture and number of parameters**, and are identical in the inference phase, as long as we add the regularization in the training of PETs.
> >
> > *Q6: Eq.5/6 uses functions not explained yet*
> >
> > A6: The *goodness-of-approx function* is specific to the fitting method, and we define the function for PDF and SDE in Sec 3.3.1 and 3.3.2 respectively. Thanks for your kind suggestions, we have improved the writing and made it clear in our rebuttal revision.
> >
> > *Q7: We do not include code and data*
> >
> > A7: **Codes are included in our supplementary material. We also included a sample of pre-training corpus and code instructions in the supplementary material.** **The** **datasets of GLUE are also available online.**

---

### Author Response · Authors · 2022-11-18
**General Response to All Reviewers**

We thank all the reviewers for your insightful suggestions! We have worked hard to polish our paper, added several essential experiments according to your valuable comments, and uploaded our rebuttal revision. The formulas and details of our methods should be much clearer in the revision. Here, we list the major modifications we have made:

**Section 2.1:**

- We add this subsection to clarify some important mathematical notations we use in the paper.

**Section 2.2:**

- We fix several problems with the notations. For example, we (1) add the subscript $o$ for $\mathbf{h}$ in the loss functions of Eq.1 and Eq.2, (2) add the previously missing PETs parameters $\phi$ in the definition of $\mathbf{h}^{(i)}$ in Eq.2.

**Section 2.3:**

- We fix mathematical notations.
- We add a brief paragraph to explain our reasons for using stochastic bridges. **(Reviewer PqBu)**

**Section 3.1:**

- We reorganize the entire subsection and elaborate our framework into three parts to make our presentation clearer.
- We add the introduction of latent states for both the PDF regularizer and the SDE regularizer. The latent states used by the PDF regularizer are with discrete time steps (Eq.5), while those used by the SDE regularizer are with continuous time (Eq.6). Eq.5 and Eq.6 will be referred to when introducing the fitting methods in Section 3.3 **(Reviewer CmgL)**
- We add an explanation for the goodness-of-approximation function when it first appears. **(Reviewer PqBu)**
- We fix notation problems and remove the subscript $o$ for $\bar{\mathbf{h}}$. **(Reviewer CmgL)**

**Section 3.3:**

- We improve the writing of the formulas and make all the functions and variables in the formulas clear. **(Reviewer CmgL)**

**Experiments:**

We place the results of additional experiments in Appendix F and G.

- Appendix F: We add the experiments when the regularizers are trained on only around 0.1% of the original pre-training corpus (10,000 documents). PETs on BERT-large with these regularizers are also greatly improved under full and few-shot settings, showing that our regularizers are data efficient. **(Reviewer 69ms)**
- Appendix G: We add the plot of time-metric curve. The regularized PETs with the PDF regulairzer are only slightly slower than vaniila PETs. The regularized PETs with the SDE regulairzer converges comparatively fast to vaniila PETs on medium-sized and small datasets, converges 2 to 3 times slower than vanilla PETs on large datasets. Considering the improvements brought by the SDE regularizer, it is very suitable for scenarios where there is no strict requirements for training time and a good PET is required to perform inference. **(Reviewer 69ms)**

---

### Decision · Program_Chairs · 2023-01-20

**Decision:**

Reject

**Justification For Why Not Higher Score:**

Concerns on multiple weaknesses are shared by reviewers, especially on the clarity of the motivation and technical presentation. The authors' feedback is not sufficient to address the concerns.

**Justification For Why Not Lower Score:**

N/A

**Metareview: Summary, Strengths And Weaknesses:**

This paper proposes a novel regularization for tuning large pre-trained language models (PLMs) with parameter-efficient tuning methods (PETs). This method is motivated with a connection of modeling intermediate state dynamics to optimal control literature and uses stochastic bridges for a target difffusion process as the regularizer. The experiments show improved performance on downstream finetuning tasks when applied to existing PETs.

Strengths:
- A novel regularization method using stochastic bridges on intermediate states rather than parameters for PET.

Weaknesses:
- Clarity. Multiple reviewers find the paper hard to follow, and the connection to optimal control unclear.
- Require original training corpus to train the regularizer which deserves more discussion than the current form.
- The significance of empirical results are not totally convincing.

**Summary Of Ac-Reviewer Meeting:**

N/A